# HYBRID FINE-TUNING OF LLMS: THEORETICAL INSIGHTS ON GENERALIZED SMOOTHNESS AND CONVERGENCE

## ABSTRACT

Applying either Parameter-Efficient Fine-Tuning (PEFT) or full fine-tuning to Large Language Models (LLMs) often results in its inherent limitations. To overcome this issue, we propose a novel "hybrid fine-tuning" approach that jointly updates both LLMs and PEFT modules using a combination of zeroth-order and first-order optimization methods. To analyze this approach, we develop a theoretical framework centered on the concept of "hybrid generalized smoothness", which accounts for the heterogeneous nature of the optimization landscape in joint LLM and PEFT training. We provide a rigorous convergence analysis for the convergence of SGD algorithm under multiple learning rates and demonstrate its effectiveness through extensive empirical studies across various downstream tasks and model architectures. Our work not only offers a solution to the practical challenge of LLM fine-tuning but also contributes a broader theoretical foundation for analyzing hybrid optimization problems in machine learning.

## 1 INTRODUCTION

Large Language Models (LLMs) have revolutionized natural language processing (NLP), demonstrating remarkable capabilities across a wide range of tasks. To adapt these models for specific domains or to modify their core behaviors, researchers and practitioners commonly employ full fine-tuning (Malladi et al., 2023; Zhang et al., 2024).

Full fine-tuning, which involves updating all parameters of an LLM, has been a classical approach for downstream tasks (VM et al., 2024; Minaee et al., 2024). However, this method is extremely computationally expensive, requiring the calculation of gradients for the entire model. To address this limitation, two common approaches have emerged: (1) *Zeroth-order full fine-tuning* (Malladi et al., 2023; Zhang et al., 2024): This method approximates gradients without directly computing them, reducing computational overhead while still allowing updates to all model parameters. (2) *Parameter-Efficient Fine-Tuning (PEFT) methods* (Lester et al., 2021; Hu et al., 2021; Li & Liang, 2021): These techniques aim to adapt LLMs by tuning only a small portion of parameters while keeping the base model frozen. This approach significantly reduces computational requirements and memory usage.

However, simply applying either of these methods has been shown to be insufficient: The PEFT method (e.g. LoRA) doesn't learn new knowledge (Gudibande et al., 2023; Ghosh et al., 2024), while the zeroth-order full-parameter fine-tuning suffers from slow convergence due to the lack of gradient information (Nesterov & Spokoiny, 2017). This limitation highlights a critical gap in current approaches, leading to the following question:

> *Q: How can we achieve both benefits of full fine-tuning and PEFT methods while maintaining the efficiency?*

To address this challenge, we propose a novel approach, *hybrid fine-tuning*, which jointly updates both the PEFT module and the LLM that adapts zeroth-order (ZO) optimization techniques to update the base model. By leveraging ZO methods, we can perform fine-tuning without calculating the full gradient of the base LLM, thereby reducing computational costs. Simultaneously, this approach

allows us to update PEFT modules using the first-order gradient information, boosting performance beyond traditional zeroth-order full fine-tuning.

However, this new approach also presents new theoretical challenges in the convergence analysis. As demonstrated in existing literature (Zhang et al., 2019; Carmon et al., 2020), the optimal learning rate is closely tied to the local smoothness of the loss landscape (i.e. the local gradient Lipschitz constant $L$ which is further detailed in Section 2.1). The complex architecture of modern large language models, combined with the heterogeneous nature of our hybrid fine-tuning approach, introduces two key theoretical challenges:

(1) **A dynamic changing gradient Lipschitz constant $L$**: The local smoothness structure of language models evolves dynamically during training. This phenomenon, first observed by Zhang et al. (2019) for LSTM-based language models, extends to transformer-based architectures, underscoring the complexity of LLM fine-tuning. Figure 1a illustrates this dynamic behavior in OPT-125M (Zhang et al., 2022), a transformer-based language model.

(2) **Heterogeneous smoothness across parameters**: The base LLM and PEFT modules exhibit distinct smoothness characteristics. Due to differences in architecture and scale, components in our proposed hybrid fine-tuning approach naturally possess diverse smoothness properties. This heterogeneity is demonstrated in Figure 1b, which compares the gradient Lipschitz constants of different modules.

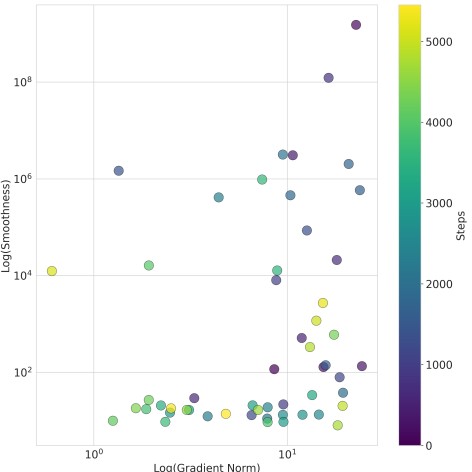

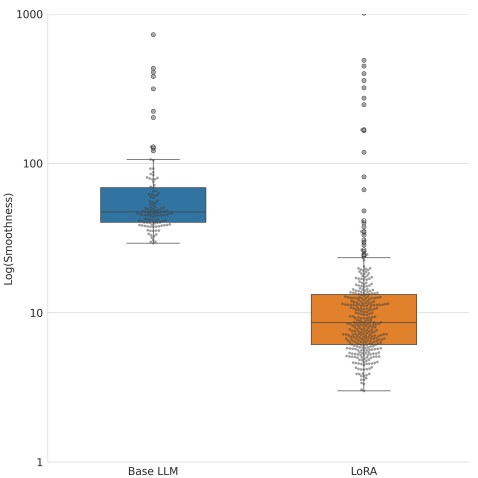

(a) Log-scale comparison of gradient norm to local gradient Lipschitz constant $L$ during OPT-125M (Zhang et al., 2022) training on the SST2 dataset (Socher et al., 2013). The colorbar indicates the number of gradient updates.

(b) Comparison of gradient Lipschitz constant $L$ for different modules (OPT-125M and LoRA (Hu et al., 2021)). The base LLM exhibits a significantly larger $L$, necessitating a smaller learning rate in the gradient updating.

Figure 1: Visualization of smoothness structures in hybrid fine-tuning a large language model. These complex characteristics pose new challenges for the convergence analysis of traditional optimization algorithms, motivating us to consider a relaxed smoothness condition, *hybrid smoothness* (Definition 2.1), for the hybrid fine-tuning method. More details are provided in Appendix D.

These challenges highlight a significant gap between existing theoretical frameworks and the practical implementation of hybrid fine-tuning methods: Traditional convergence analysis of optimization algorithms cannot be applicable for such complicated loss surface, which also leads to the following central question:

> Q: *How can we develop a unified theoretical framework that accurately characterizes the convergence of SGD for hybrid fine-tuning while accounting for their distinct characteristics and behaviors?*

To answer this question, we develop a novel theoretical framework centered around the concept of "hybrid generalized smoothness". This framework provides a more accurate characterization of the

optimization landscape in joint LLM and PEFT training, enabling rigorous analysis of convergence properties and optimization dynamics.

## 1.1 CONTRIBUTIONS

We summarize our contributions as follows:

(1) We introduce the hybrid fine-tuning paradigm, a novel approach that addresses the limitations of both full fine-tuning and traditional PEFT methods. By combining zeroth-order optimization for LLMs with first-order methods for PEFT modules, we achieve a balance between adaptation power and computational efficiency. This innovative strategy allows for more comprehensive model updates without incurring the full computational cost of traditional fine-tuning.

(2) We develop a comprehensive theoretical framework for analyzing hybrid optimization problems in the context of hybrid fine-tuning. Our concept of *hybrid generalized smoothness* (Definition 2.1) extends classical optimization theory to account for the heterogeneous nature of joint LLM and PEFT training. We provide rigorous convergence analysis for our proposed algorithm under this new relaxed smoothness condition, establishing a solid theoretical foundation for hybrid fine-tuning approaches. Notably, we analyze SGD with random reshuffling, a more common variant used in practice. Furthermore, we demonstrate the broad applicability of our theoretical framework beyond hybrid fine-tuning, providing insights into various LLM applications such as layer-wise fine-tuning (Zhang et al., 2024) or models with trainable external modules (Raissi et al., 2019). This generalization enhances the impact and utility of our theoretical contributions across diverse machine learning paradigms involving heterogeneous parameter spaces.

(3) We present an extensive empirical study demonstrating the effectiveness of hybrid fine-tuning across a diverse range of downstream tasks and model architectures. We observe consistent gains over traditional PEFT techniques and zeroth-order full fine-tuning. Our results not only validate the theoretical insights but also showcase significant improvements in adaptation quality, computational efficiency, and model performance compared to existing methods.

By addressing the fundamental challenges of joint LLM and PEFT training, our work opens new avenues for research in efficient model adaptation in the context of large-scale language models. The theoretical framework we propose has the potential to serve as a new foundation for analyzing and optimizing hybrid systems across a broad range of applications and domains, extending its impact beyond the specific context of language model fine-tuning.

## 1.2 RELATED WORK

**Zeroth-order Optimization in Fine-tuning LLMs**    Recent work has explored zeroth-order optimization methods for fine-tuning LLMs, which aligns with our approach of using zeroth-order methods for the LLM component in hybrid fine-tuning. Malladi et al. (2023) demonstrated the compatibility of zeroth-order methods with both full fine-tuning and PEFTs. This laid the groundwork for our hybrid approach that combines zeroth-order LLM updates with first-order PEFT updates. Zhang et al. (2024) provided a comprehensive benchmark for zeroth-order optimization in LLM fine-tuning, offering valuable insights that informed our experimental design. Ling et al. (2024) combines the zeroth-order fine-tuning of LLMs with the federated learning. Several studies have incorporated variance reduction techniques (Gautam et al., 2024) into zeroth-order methods or second-order method (Zhao et al., 2024) to enhance stability and convergence in fine-tuning LLMs. While we focus on a different aspect, these stability improvements could easily be integrated into our hybrid framework. Existing literature (Liu et al., 2024; Guo et al., 2024; Zhang et al., 2024) also discusses the sparsity of pre-trained LLMs, which further enhances the performance of zeroth-order optimization approach.

**Generalized Smoothness of Large Machine Learning Models**    The concept of generalized smoothness has emerged as a crucial theoretical framework for understanding the optimization landscape of large machine learning models, including LLMs. Recent studies have shown that traditional smoothness assumptions often fail to capture the complex optimization landscape of deep neural networks (Zhang et al., 2019; Li et al., 2024). More explicitly, Zhang et al. (2019) demonstrated that the

local smoothness constant in neural networks is often proportional to the gradient norm, challenging the conventional assumption of uniform smoothness. This insight aligns with our observations in hybrid fine-tuning, where different components of the model (LLM and PEFT modules) exhibit distinct smoothness properties. Li et al. (2024) introduced a generalized smoothness condition that allows for non-uniform smoothness across the parameter space, which is more representative of the behavior observed in practice for large models. This work provides a foundation for our hybrid generalized smoothness framework, which extends these ideas to account for the heterogeneous nature of joint LLM and PEFT optimization.

## 2 THE HYBRID SMOOTHNESS CONDITION FOR HYBRID SYSTEMS

In this section, we further described the theoretical challenges in jointly training both PEFT modules and the base LLM. We abstractizare the fine tuning of the LLM with multiple modules as a class of optimization problems where the parameter space is partitioned into two distinct subsets, each exhibiting different smoothness properties. It is formally described as follows:

$$\min_{(x,y)\in\mathbb{R}^d} f(x,y) := \frac{1}{n}\sum_{i=1}^{n} f(x,y;i). \tag{1}$$

Here, $x \in \mathbb{R}^{d_x}$ and $y \in \mathbb{R}^{d_y}$ are the parameters of the model, with $d = d_x + d_y$. The objective function $f : \mathbb{R}^d \to \mathbb{R}$ is typically a loss function in the context of machine learning tasks. The index $i$ represents individual data points in a dataset of size $n$. Importantly, we consider the scenario where the gradient of $f$ with respect to $x$ is not directly accessible due to the memory issue. This reflects the practical constraints often encountered in fine-tuning large language models, which is commonly solved using zeroth-order optimization approach to relax the memory constraints (Malladi et al., 2023; Gautam et al., 2024; Ling et al., 2024; Guo et al., 2024; Liu et al., 2024; Zhang et al., 2024).

### 2.1 THE $L$-SMOOTHNESS CONDITION

In the traditional optimization problem, which is usually used to characterize the full fine-tuning of LLM or the PEFT methods, the concept of $L$-smoothness is fundamental in the optimization theory and plays a crucial role in characterizing the behavior of SGD algorithms.

Formally, a smooth function $f : \mathbb{R}^d \to \mathbb{R}$ is said to be $L$-*smooth* (also known as $L$-*Lipschitz continuous gradient*) if there exists a constant $L > 0$ such that its Hessian matrix is uniformly bounded by the constant $L$; that is,

$$LI_d \succeq \nabla^2 f(x),$$

where $I_d$ is the $d \times d$ identity matrix and "$\succeq$" represents that $LI_d - \nabla^2 f(x)$ is positive semi-definite (PSD). This condition can be equivalently expressed in terms of the function's gradient: $\|\nabla f(x) - \nabla f(y)\| \le L\|x - y\|$. The constant $L$ is closely related to the learning rate choice in the convergence analysis of gradient-based algorithm. As demonstrated in Carmon et al. (2020), the optimal learning rate is linearly scaled with respect to $\frac{1}{L}$.

While $L$-smoothness has been demonstrated to hold for all smooth functions over a compact domain (Hewitt & Stromberg, 2012), it has limitations when applied to complicated landscapes encountered in our hybrid training approach described in Eq. (1). We recap these limitations we have introduced:

(1) $L$ **is dynamically changing during training**. The constant $L$ usually fails to maintain uniformity over the entire parameter space. In many practical scenarios, different regions of the parameter space may exhibit vastly different smoothness properties. For instance, Zhang et al. (2019) has demonstrated that the local smoothness constant $L$ is linear in the gradient norm. We also have illustrated this non-uniformity for transformer-based language models in Figure 1a.

(2) $L$ **can be different for different parameters**. Distinct types of modules or variables within the system usually present highly diverse smoothness conditions. For example, small randomly-initialized modules often have smaller $L$ compared to large pre-trained neural networks. This consideration becomes particularly crucial in hybrid systems where we deal with fundamentally different types of parameters. We have illustrated this point in Figure 1b: The LoRA module demonstrates a substantially lower $L$ value compared to the base LLM.

## 2.2 Hybrid Generalized Smoothness Condition

To address the limitations of traditional $L$-smoothness in our hybrid optimization framework given in Eq. (1), we introduce the concept of hybrid generalized smoothness:

**Definition 2.1** (Hybrid generalized smoothness). A function $f : \mathbb{R}^{d_x} \times \mathbb{R}^{d_y} \to \mathbb{R}$ has the hybrid generalized smoothness property if there exist two non-negative non-decreasing sub-quadratic function $\ell_x : \mathbb{R} \to \mathbb{R}$ and $\ell_y : \mathbb{R} \to \mathbb{R}$ such that:

$$\begin{bmatrix} \ell_x(\|\nabla f(x,y)\|)I_{d_x} & 0 \\ 0 & \ell_y(\|\nabla f(x,y)\|)I_{d_y} \end{bmatrix} \succeq \nabla^2 f(x,y).$$

This definition extends the generalized smoothness in Li et al. (2024) and allows for different smoothness properties in different parts of the parameter space, represented by $x$ and $y$. The functions $\ell_x$ and $\ell_y$ can capture varying degrees of smoothness for different types of parameters. This definition more accurately characterizes the loss surface of our proposed hybrid fine-tuning method. The following proposition demonstrates that the classical $L$-smoothness is indeed a stronger condition than hybrid generalized smoothness.

**Proposition 2.2** (L-smoothness implies hybrid generalized smoothness). *If a function $f : \mathbb{R}^{d_x} \times \mathbb{R}^{d_y} \to \mathbb{R}$ is L-smooth, then it satisfies the hybrid generalized smoothness condition with $\ell_x(t) = \ell_y(t) = L$ for all $t \geq 0$.*

*Proof.* For an $L$-smooth function, we have $\nabla^2 f(x,y) \preceq LI_{d_x+d_y}$ for all $(x,y)$. This implies:

$$\begin{bmatrix} LI_{d_x} & 0 \\ 0 & LI_{d_y} \end{bmatrix} \succeq \nabla^2 f(x,y).$$

By setting $\ell_x(t) = \ell_y(t) = L$ for all $t \geq 0$, we satisfy the condition in Definition 2.1, thus proving that $L$-smoothness implies hybrid generalized smoothness. □

It is also worth noting that most standard neural network structures have been empirically verified as generalized smooth, but not $L$-smooth for any constant $L$ (Zhang et al., 2019; Li et al., 2024). The hybrid generalized smoothness condition allows for more flexibility in capturing the smoothness properties of complex optimization landscapes, particularly those encountered in training hybrid systems.

## 2.3 The Impact of Hybrid Generalized Smoothness

The concept of hybrid generalized smoothness has significant implications for optimization strategies, particularly motivating the use of two distinct learning rates. In this section, we will provide an intuitive explanation on the necessity of applying two learning rates in optimizing Eq. (1) for improving the efficiency when the hybrid generalized smoothness presents.

To illustrate the impact and utility of this approach, let's consider two examples:

**Example 2.3** (Quadratic Function). Consider a simple quadratic function $f(x,y) = ax^2 + by^2$, where $a$ and $b$ are positive constants, and $a \gg b$. The gradient is $\nabla f(x,y) = (2ax, 2by)$, and the Hessian is:

$$\nabla^2 f(x,y) = \begin{bmatrix} 2a & 0 \\ 0 & 2b \end{bmatrix}.$$

If we use a single learning rate $\eta$ for both $x$ and $y$, we would typically choose $\eta$ based on the largest eigenvalue of the Hessian to ensure stability: $\eta = \frac{1}{2\max(a,b)} = \frac{1}{2a}$. The update rules would then be:

$$\begin{bmatrix} x_{t+1} \\ y_{t+1} \end{bmatrix} = \begin{bmatrix} 1 - 2a\eta & 0 \\ 0 & 1 - 2b\eta \end{bmatrix} \begin{bmatrix} x_t \\ y_t \end{bmatrix} = \begin{bmatrix} 0 & 0 \\ 0 & 1 - \frac{b}{a} \end{bmatrix} \begin{bmatrix} x_t \\ y_t \end{bmatrix}.$$

This leads to an issue: The $y$ component converges to the optimal value $y^* = 0$ very slowly, as $\frac{b}{a} \ll 1$, resulting in minimal updates to $y$ in each iteration.

In the contrast, if we increase the learning rate $\eta$ to $\frac{1}{2b}$, we of course obtain a much faster convergence at the component $y$. However, the update rule leads to

$$\begin{bmatrix} x_{t+1} \\ y_{t+1} \end{bmatrix} = \begin{bmatrix} 1 - 2a\eta & 0 \\ 0 & 1 - 2b\eta \end{bmatrix} \begin{bmatrix} x_t \\ y_t \end{bmatrix} = \begin{bmatrix} 1 - \frac{a}{b} & 0 \\ 0 & 0 \end{bmatrix} \begin{bmatrix} x_t \\ y_t \end{bmatrix}.$$

The $x$ component simply diverges to infinity because $\frac{a}{b} \gg 1$.

When choosing different learning rates for $x$ and $y$, we set $\eta_x = \frac{1}{2a}$ and $\eta_y = \frac{1}{2b}$. Then the update rules become:

$$x_{t+1} = x_t - \eta_x \cdot 2ax_t = x_t - x_t = 0,$$
$$y_{t+1} = y_t - \eta_y \cdot 2by_t = y_t - y_t = 0.$$

With these tailored learning rates, both $x$ and $y$ converge at similar rates (in this case, in a single step), despite the significant difference in their quadratic coefficients.

This simple example demonstrates our motivation of choosing different learning rates when facing the hybrid generalized smoothness. The similar phenomenon is also observed in our proposed hybrid LLM fine-tuning structure:

**Example 2.4** (Fine-Tuning a LLM with PEFT Modules)**.** Consider hybrid fine-tuning a base LLM with a PEFT module. In this scenario, we have two sets of parameters:

- $x$: The original LLM parameters.

- $y$: The PEFT module parameters (e.g., LoRA or adapters).

In this example, we jointly train the LLM with a Prompt Encoder on the SST-2 dataset. Due to the memory limitation, it is common to restrict the LLM update using the zeroth-order optimization approach. We observe that the base LLM merely takes much smaller learning rate; if we choose the learning rate to ensure the base LLM's convergence, the training loss decreases in an unaccetably slow rate (Figure 2a). However, if we choose the learning rate larger than the base LLM's tolerance, the training loss explodes and quickly diverges (Figure 2b). The best practice is choosing a smaller learning rate for the base LLM and a larger learning rate for the PEFT module (Figure 2c).

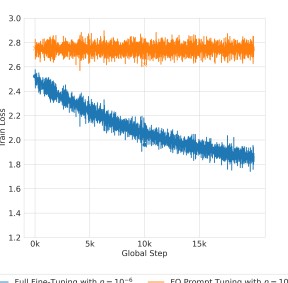 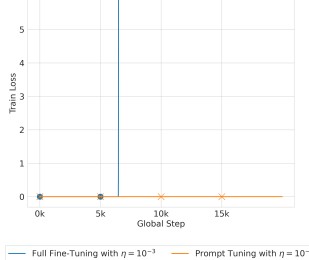 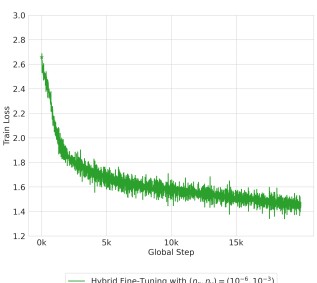

(a) Under the small learning rate $\eta = 10^{-6}$, the full fine-tuning decrease as expected. However, prompt tuning converges slowly and stagnates at a higher loss.

(b) With a large learning rate $\eta = 10^{-3}$, the prompt tuning decreases as expected, while full fine-tuning exhibits unstable behavior, resulting in loss explosion.

(c) Hybrid fine-tuning with distinct learning rates ($\eta_x = 10^{-6}$ for the base model, $\eta_y = 10^{-3}$ for prompt tuning) provides both stable and faster convergence.

Figure 2: Comparison of training loss curves under different learning rate configurations for full fine-tuning and prompt tuning on the SST-2 dataset (Socher et al., 2013) with the base model OPT-1.3b (Zhang et al., 2022). This example illustrates the necessaity of using different learning rates in hybrid-tuning structure.

These examples illustrate the practical benefits of considering hybrid generalized smoothness and the resulting use of multiple learning rates in training a complex hybrid system. By tailoring the optimization process to the specific smoothness of different components, we can potentially achieve faster convergence and better overall performance. However, to fully understand the implications and guarantees of this approach, we need to delve deeper into its theoretical foundations.

# 3 THEORETICAL ANALYSIS

In this section, we aim to bridge the gap between the practical benefits of generalized hybrid smoothness and its underdeveloped theoretical foundations. We present a rigorous analysis of the Stochastic Gradient Descent (SGD) algorithm with random reshuffling. The random reshuffling is widely used in practice, particularly in NLP problems. The algorithm we consider is described in Algorithm 1.

---

**Algorithm 1:** SGD with Random Reshuffling for Hybrid Fine-Tuning

**Input:** Learning rate $\eta = \begin{bmatrix} \eta_x & \eta_y \end{bmatrix}$, number of epochs $T$, dataset $\mathcal{D} = \{\xi_i\}_{i=1}^n$

1 Initialize the parameter at $(x_0, y_0)$;
2 **for** $t = 1$ **to** $T$ **do**
3      Shuffle the dataset $\mathcal{D}$ to obtain $\mathcal{D}_t$;
4      $\begin{bmatrix} x_{t,0} \\ y_{t,0} \end{bmatrix} \leftarrow \begin{bmatrix} x_{t-1} \\ y_{t-1} \end{bmatrix}$;
5      **for** $i = 1$ **to** $n$ **do**
6          $\begin{bmatrix} x_{t,i} \\ y_{t,i} \end{bmatrix} \leftarrow \begin{bmatrix} x_{t,i-1} \\ y_{t,i-1} \end{bmatrix} - \begin{bmatrix} \eta_x & 0 \\ 0 & \eta_y \end{bmatrix} \begin{bmatrix} \hat{\nabla}_x f(x_{t,i}, y_{t,i}; \xi_{t,i}) \\ \nabla_y f(x_{t,i}, y_{t,i}; \xi_{t,i}) \end{bmatrix}$;
7      **end**
8 **end**
9 $x_t \leftarrow x_{t,n}$;
**Output:** Final parameters $x_T$

---

In this algorithm, $\eta_x$ and $\eta_y$ are the learning rates for $x$ and $y$ parameters, respectively, $T$ is the total number of epochs, $\mathcal{D} = \{\xi_i\}_{i=1}^n$ is the dataset with $n$ samples, and $x_{t,i}$ and $y_{t,i}$ are the parameter values after the $i$-th iteration of the $t$-th epoch. $\hat{\nabla}_x f$ and $\nabla_y f$ are the stochastic gradients with respect to $x$ and $y$. Here, we use $\hat{\nabla}_x f$ to represent the gradient estimator of $\nabla_x f$. It is commonly estimated using either one-side or two-side gradient estimator defined as follows:

$$\hat{\nabla}_x f(x, y) := \frac{f(x + \mu v, y) - f(x, y)}{\mu} v, \tag{2}$$

where $v$ is a random vector sampled from the Gaussian distribution $N(0, I_d)$ and $\mu$ is the perturbation stepsize.

We emphasize that while analysis of step-wise SGD (i.e., randomly sampling one data point from the dataset) is more straightforward, epoch-wise updates are typically more common in machine learning practice. Moreover, existing literature (Ma & Zhou, 2020; Safran & Shamir, 2020; Mishchenko et al., 2020; Gürbüzbalaban et al., 2021; Liu & Zhou, 2024) has demonstrated that random reshuffling can improve the efficiency of the SGD algorithm under certain conditions. To increase the practical relevance of our theory, we focus on this epoch-wise update rule with random reshuffling.

## 3.1 PROBLEM FORMULATION AND ASSUMPTIONS

Our objective is to solve the optimization problem presented in Eq. (1). As justified in the previous section, we assume the objective function $f : \mathbb{R}^{d_x} \times \mathbb{R}^{d_y} \to \mathbb{R}$ satisfies the hybrid generalized smoothness (Definition 2.1). To handle the generalized smooth structure, we introduce the following definition:

**Definition 3.1** (Coercive). A continuous function $f : \mathbb{R}^d \to \mathbb{R}$ is *coercive* if the sub-level set $\{x \in \mathbb{R}^d \mid f(x) \le a\}$ is compact for all $a \in \mathbb{R}$.

In the existing literature of generalized smoothness (Li et al., 2024), this assumption is usually replaced with an equivalent statement: the objective function $f(x, y)$ tends to positive infinity when $(x, y)$ approaches the boundary of its domain. In addition to the assumption, we make the following standard assumptions to regularize the function class and subsequently provide the non-asymptotic convergence analysis.

**Assumption 3.2** (Regularity Conditions). *The objective function* $f(x, y) := \frac{1}{n} \sum_{i=1}^n f(x, y; i)$ *defined in Eq. (1) satisfies the following conditions:*

*(1) $f(\cdot)$ is coercive.*

*(2) $f(\cdot)$ is bounded below by $f^* := \inf_{(x,y) \in \mathbb{R}^d} f(x,y) > -\infty$.*

*(3) $f(\cdot)$ and each individual loss function $f(\cdot;i)$ are twice continuously differentiable.*

These regularity conditions are essential for several reasons: Coercivity prevents the optimization process from diverging too far. The lower bound guarantees that the optimization problem is well-posed. Twice continuous differentiability allows for the application of various optimization techniques and facilitates theoretical analysis. All of them are standard and widely used in the optimization literature (Li et al., 2024).

**Assumption 3.3** (Bounded variance)**.** *There exists $\sigma$ such that for all $x \in \mathbb{R}^d$,*

$$\frac{1}{n} \sum_{i=1}^{n} \|\nabla f(x,y;i) - \nabla f(x,y)\|^2 \le \sigma^2.$$

This bounded variance assumption is standard in the analysis of reshuffling-type SGD. We note that this assumption could be further weakened to the expected smoothness (Mishchenko et al., 2020; Khaled & Richtárik, 2020). We maintain the current version for the simplicity.

These assumptions collectively provide the necessary foundation for our subsequent analysis, allowing us to derive meaningful convergence guarantees for the SGD algorithm in the context of hybrid fine-tuning with generalized smoothness.

### 3.2 NON-ASYMPTOTIC CONVERGENCE ANALYSIS

In this section, we analyze the complexity of Algorithm 1 under under our hybrid generalized smoothness condition. Our main theoretical result is summarized in the following theorem:

**Theorem 3.4.** *Suppose that Assumption 3.2 and Assumption 3.3 hold for the objective function $f(x,y) := \frac{1}{n} \sum_{i=1}^{n} f(x,y;i)$, with satisfying the hybrid generalized smoothness property (Definition 2.1). Let $\{(x_t, y_t)\}_{t=1}^{T}$ be the SGD dynamic generated by Algorithm 1 for solving the optimization problem Eq. (1). Let learning rates $\eta_x$, $\eta_y$ be chosen as*

$$\eta_x \le \min \left\{ \mathcal{O}(\frac{1}{L_x n d_x}), \mathcal{O}(\frac{1}{\sqrt{T} n L_{x,\max}}) \right\},$$

$$\eta_y \le \min \left\{ \mathcal{O}(\frac{1}{L_y n}), \mathcal{O}(\frac{1}{\sqrt{T} n L_{y,\max}}) \right\},$$

*and the perturbation stepsize $\mu$ is specified as Eq. (3). Here, $L_x, L_y, L_{x,\max}, L_{y,\max}$ are specified in Appendix C, representing the smoothness characteristics of the $x$ and $y$ parameters, respectively. Let $\delta \in (0,1)$. If the maximum number of epoch $T$ is chosen as $T \ge \mathcal{O}(\frac{\epsilon^{-2}}{\delta} + \frac{\epsilon^{-4}}{n})$, then with the probability at least $1 - \delta$,*

$$\frac{1}{T} \sum_{t<T} \mathbf{E} \|\nabla f(x_t, y_t)\|^2 \le \epsilon^2.$$

*Proof.* The full version and the proof are deferred to Appendix C. □

Given that each epoch processes $n$ data points, the total gradient complexity is $nT \ge \mathcal{O}(\frac{\epsilon^{-2}n}{\delta} + \epsilon^{-4})$. This result is optimal when $\epsilon$ is sufficiently small, aligning with the best-known upper bounds established in previous convergence analyses for both generalized smooth non-convex objectives (Li et al., 2024; Zhang et al., 2019) and $L$-smooth non-convex objectives (Mishchenko et al., 2020; Khaled & Richtárik, 2020). Importantly, it also matches the known lower bound for the SGD algorithm (Arjevani et al., 2023), further confirming its optimality. Our analysis yields several important insights:

(1) Our analysis reveals the asymmetry between the learning rates $\eta_x$ and $\eta_y$, which is closely tied to the smoothness properties of each variable. This finding underscores the importance of tailored learning rate configurations when modules exhibit diverse smoothness characteristics.

Table 1: Experiment results for various fine-tuning methods applied to three large language models (Llama-2-7b, Vicuna-7b-v1.5, and OPT-1.3b) across three NLP tasks (SST2, Copa, and Wino-Grande). Highlighted cells indicate the best performance achieved by the hybrid-tuning approach for each model-task combination.

|  |  | SST2 | Copa | WinoGrande |
|---|---|---|---|---|
| | ZO-FT | 93.58 | 87 | 67.5 |
| | FO-Prompt | 95.64 | 88 | 67.2 |
| | Hybrid-Prompt | 95.9 | 88 | 68.9 |
| Llama-2-7b | FO-Prefix | 91.05 | 83 | 66.2 |
| | Hybrid-Prefix | 91.63 | 85 | 64.3 |
| | FO-Lora | 94.61 | 84 | 68.5 |
| | Hybrid-Lora | 93.4 | 88 | 66.3 |
| | ZO-FT | 91.40 | 87 | 64.7 |
| | FO-Prompt | 94.38 | 84 | 65.8 |
| | Hybrid-Prompt | 94.95 | 84 | 66.3 |
| Vicuna-7b-v1.5 | FO-Prefix | 90.02 | 80 | 64.1 |
| | Hybrid-Prefix | 90.71 | 83 | 74 |
| | FO-Lora | 94.61 | 85 | 66.7 |
| | Hybrid-Lora | 92.20 | 84 | 66.7 |
| | ZO-FT | 91.51 | 78 | 57.9 |
| | FO-Prompt | 91.28 | 74 | 57.8 |
| | Hybrid-Prompt | 91.74 | 77 | 59.9 |
| OPT-1.3b | FO-Lora | 92.2 | 78 | 59 |
| | Hybrid-Lora | 92.3 | 78 | 58.3 |
| | FO-prefix | 92.2 | 77 | 58.3 |
| | Hybrid-Prefix | 91.7 | 78 | 60 |

The gradient estimation process further accentuates the asymmetry in learning rate selection. For the $x$ parameter, which is updated using zeroth-order gradient estimation, the learning rate incorporates an additional scaling factor of $\frac{1}{d_x}$, where $d_x$ represents the dimensionality of the $x$ parameter space. This theoretical insight aligns with our empirical observations: in practice, we find that the learning rate for updating the LLM is typically much smaller than the learning rate used for PEFT modules. This correlation between theory and practice not only validates our analytical framework but also provides valuable guidance for hyperparameter tuning in hybrid LLM systems.

(2) Compared to Li et al. (2024), our derived sample complexity of $nT \geq \mathcal{O}(\frac{n\epsilon^{-2}}{\delta} + \epsilon^{-4})$ represents an improvement in the dependence of $\delta$ over existing results in the literature $\mathcal{O}(\frac{\epsilon^{-4}}{\delta})$. This improvement is made by using a stronger version of concentration inequality, replacing the Markov inequality used in its original proof. Notably, this technical improvement has potential applications beyond our specific setting, extending to other optimization algorithms under generalized smoothness conditions.

## 4 EXPERIMENTS

We conducted extensive experiments to evaluate the effectiveness of our proposed hybrid fine-tuning approach across various tasks, model architectures, and PEFT methods.

**Experiment Details** Following the methodology of Malladi et al. (2023), we assessed our approach on several NLP tasks, including the sentiment classification task on the SST2 dataset (Socher et al., 2013), the question answering task on the COPA dataset (Roemmele et al., 2011), and the common sense reasoning task on the WinoGrande dataset (Sakaguchi et al., 2021). For each dataset, we randomly sample 1,000 examples for training, 1,000 examples for evaluation, and 100 examples for development. The models we use in our experiments include OPT-1.3b (Zhang et al., 2022), Vicuna-7b (Chiang et al., 2023), and LLaMA-7b (Zhang et al., 2023b). We compare the performance of our approach against several methods: zeroth-order full model fine-tuning as described in

Table 2: A detailed breakdown of the optimal hyperparameters including learning rates, training steps and $\mu$ specified in Eq. (2) and training specifics for each fine-tuning method applied to different model architectures across various NLP tasks. Highlighted cells indicate efficient training processes, showcasing the reduced steps required by hybrid approaches to achieve optimal performance.

| | | SST2 | | | Copa | | | WinoGrande | | |
| --- | --- | --- | --- | --- | --- | --- | --- | --- | --- | --- |
| | | learning rate (PEFT/Base) | steps | $\mu$ | learning rate (PEFT/Base) | steps | $\mu$ | learning rate(s) (PEFT/Base) | steps | $\mu$ |
| Llama-2-7b | ZO-FT | $10^{-6}$ | $1.1*10^4$ | $10^{-5}$ | $10^{-6}$ | $1.6*10^4$ | $10^{-4}$ | $10^{-6}$ | $1.8*10^4$ | $10^{-5}$ |
| | FO-Prompt | $10^{-3}$ | $6*10^3$ | / | $10^{-4}$ | $9*10^3$ | / | $10^{-3}$ | $9*10^3$ | / |
| | Hybrid | $10^{-3}/10^{-8}$ | $1.5*10^3$ | $10^{-5}$ | $10^{-4}/10^{-8}$ | $5*10^3$ | $10^{-5}$ | $10^{-3}/10^{-7}$ | $9*10^3$ | $10^{-5}$ |
| | FO-Prefix | $10^{-3}$ | $2*10^4$ | / | $10^{-3}$ | $1.5*10^4$ | / | $10^{-2}$ | $3*10^3$ | / |
| | Hybrid-Prefix | $10^{-3}/10^{-3}$ | $9.5*10^3$ | $10^{-5}$ | $10^{-3}/10^{-6}$ | $7.5*10^3$ | $10^{-5}$ | $10^{-3}/10^{-6}$ | $9*10^3$ | $10^{-5}$ |
| | FO-Lora | $10^{-4}$ | $2*10^4$ | / | $10^{-3}$ | $2.5*10^3$ | / | $10^{-2}$ | $2.5*10^3$ | / |
| | Hybrid-Lora | $10^{-4}/10^{-7}$ | $1.6*10^4$ | $10^{-5}$ | $10^{-4}/10^{-7}$ | $1.15*10^4$ | $10^{-5}$ | $10^{-3}/10^{-6}$ | $4.5*10^3$ | $10^{-5}$ |
| Vicuna-7b-v1.5 | ZO-FT | $10^{-6}$ | $10^4$ | $10^{-5}$ | $10^{-6}$ | $7*10^3$ | $10^{-5}$ | $10^{-6}$ | $1.75*10^4$ | $10^{-5}$ |
| | FO-Prompt | $10^{-3}$ | $2*10^4$ | / | $10^{-3}$ | $1.3*10^4$ | / | $10^{-3}$ | $2*10^4$ | / |
| | Hybrid | $10^{-3}/10^{-7}$ | $2*10^3$ | $10^{-5}$ | $10^{-4}/10^{-8}$ | $1.5*10^3$ | $10^{-5}$ | $10^{-3}/10^{-8}$ | $2*10^4$ | $10^{-6}$ |
| | FO-Prefix | $10^{-3}$ | $2*10^3$ | / | $10^{-2}$ | $2*10^4$ | / | $10^{-3}$ | $2*10^4$ | / |
| | Hybrid-Prefix | $10^{-3}/10^{-6}$ | $2*10^4$ | $10^{-5}$ | $5*10^{-4}/5*10^{-7}$ | $1.7*10^4$ | $10^{-5}$ | $10^{-3}/10^{-3}$ | $4*10^3$ | $10^{-5}$ |
| | FO-Lora | $10^{-3}$ | $2*10^3$ | / | $10^{-2}$ | $9*10^3$ | / | $10^{-3}$ | $3.5*10^3$ | / |
| | Hybrid-Lora | $10^{-3}/10^{-6}$ | $2*10^4$ | $10^{-5}$ | $10^{-4}/10^{-7}$ | $2.5*10^3$ | $10^{-5}$ | $10^{-3}/10^{-6}$ | $3*10^3$ | $10^{-5}$ |
| OPT-1.3b | ZO-FT | $10^{-7}$ | $2*10^4$ | $10^{-5}$ | $10^{-6}$ | $8.5*10^3$ | $10^{-4}$ | $10^{-7}$ | $8*10^3$ | $10^{-5}$ |
| | FO-Prompt | $10^{-3}$ | $2*10^4$ | / | $10^{-4}$ | $1.6*10^4$ | / | $10^{-3}$ | $9.5*10^3$ | / |
| | Hybrid-Prompt | $10^{-3}/10^{-7}$ | $2*10^4$ | $10^{-5}$ | $10^{-3}/10^{-7}$ | $2*10^4$ | $10^{-5}$ | $10^{-3}/10^{-7}$ | $1.4*10^4$ | $10^{-5}$ |
| | FO-Lora | $10^{-3}$ | $3*10^4$ | / | $10^{-4}$ | $1.9*10^4$ | / | $10^{-4}$ | $1.45*10^4$ | / |
| | Hybrid-Lora | $10^{-4}/7*10^{-10}$ | $4*10^3$ | $10^{-5}$ | $10^{-5}/10^{-11}$ | $1.9*10^4$ | $10^{-5}$ | $5*10^{-4}/5*10^{-4}$ | $3*10^3$ | $5*10^{-4}$ |
| | FO-prefix | $10^{-2}$ | $2*10^4$ | / | $5*10^{-3}$ | $2*10^4$ | / | $5*10^{-2}$ | $9.5*10^3$ | / |
| | Hybrid-Prefix | $8*10^{-3}/8*10^{-5}$ | $8.5*10^3$ | $10^{-5}$ | $2*10^{-3}/10^{-7}$ | $1.15*10^4$ | $10^{-5}$ | $5*10^{-2}/10^{-4}$ | $2*10^4$ | $10^{-5}$ |

Malladi et al. (2023), first-order prompt tuning (Lester et al., 2021), LoRA tuning (Hu et al., 2021), and prefix tuning (Li & Liang, 2021). Detailed overviews of the tasks and PEFT methods are provided in Appendix D.1 and Appendix D.2, respectively. Performance was evaluated using accuracy or F1 score, as appropriate for each task. For the zeroth-order approximation, we follow the same approach outlined by Malladi et al. (2023). All experiments utilize SGD as the optimizer. In the case of prompt tuning and prefix tuning, the prompts are initialized according to the predefined settings in Table E.2 of Malladi et al. (2023), while for LoRA tuning, we initialize with zeros. We perform hyperparameter tuning for all methods and report the best configurations. Learning rates for each method are summarized in Table 2. For all methods, we set the maximum number of training steps to 20,000, with early stopping applied when applicable.

**Results** Table 1 presents the outcomes of all experiments. The results show that, in most cases, the hybrid method outperforms zeroth-order fine-tuning and its corresponding first-order PEFT. Additionally, as shown in Table 2, the number of steps required by the hybrid method to achieve optimal performance is significantly lower than that of either zeroth-order fine-tuning or its corresponding first-order PEFT. These findings suggest that hybrid tuning offers a more efficient and effective approach to fine-tuning large language models for downstream tasks. As a supplementary, we further the efficiency of the hybrid-tuning approach in Appendix D.3.

## 5 CONCLUSION

In conclusion, this work introduces a novel hybrid fine-tuning approach for large language models that combines zeroth-order optimization for the base model with first-order optimization for PEFT modules. Motivated by the hybrid generalized smoothness of the hybrid system in Section 2.1, we develop a theoretical framework centered on this theoretical challenge introduced by the hybrid fine-tuning method. Our empirical examples and convergence analysis built in Theorem 3.4 demonstrate the necessity of applying different learning rates for different PEFT modules. Our analysis achieves the best-known sample complexity under much milder conditions in the existing literature. Extensive empirical evaluations across multiple NLP tasks, model architectures, and PEFT techniques validate the theoretical insights and show consistent performance gains over traditional fine-tuning methods. By addressing fundamental challenges in joint LLM and PEFT training, our work opens new avenues for efficient model adaptation and provides a solid foundation for future research on optimizing hybrid systems in machine learning.

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

## A  Notations

In this paper, the optimization problem is formulated as minimizing $f(x, y)$, where $x \in \mathbb{R}^{d_x}$ represents the parameters of the base language model and $y \in \mathbb{R}^{d_y}$ represents the parameters of the PEFT module. The function $f$ is assumed to have hybrid generalized smoothness, characterized by non-negative, non-decreasing sub-quadratic functions $\ell_x$ and $\ell_y$ (Definition 2.1). In the SGD, we consider epoch-wise optimization algorithm described in Algorithm 1. This approach ensures us to access each data point exactly once over an entire epoch, which is particularly common is the data loader provided by existing modern machine learning frameworks such as PyTorch and TensorFlow. Here, $\eta_x$ and $\eta_y$ denote the learning rates for $x$ and $y$ respectively, $T$ is the number of epochs, and $n$ is the dataset size. We $\hat{\nabla}_x f$ to denote the zeroth-order gradient estimator for $x$, while $\nabla_y f$ represents the standard gradient for $y$. With these given, for each epoch $t$, we define the following notations:

$$g_t = \sum_{i=1}^{n} \nabla_x f(x_{t,i}, y_{t,i}; \xi_{t,i}), \quad \hat{g}_t = \sum_{i=1}^{n} \hat{\nabla}_x f(x_{t,i}, y_{t,i}; \xi_{t,i}),$$

$$h_t = \sum_{i=1}^{n} \nabla_y f(x_{t,i}, y_{t,i}; \xi_{t,i}).$$

Here, $g_t$ represents the true gradient with respect to $x$ accumulated over an entire epoch. It captures the overall direction of stochastic gradient descent for the $x$ parameters across all samples in the epoch. $\hat{g}_t$ is an estimate of this gradient. In practice, we often don't have access to the true gradient and must rely on estimates. The difference between $g_t$ and $\hat{g}_t$ quantifies the estimation error in our gradient calculations. $h_t$ is the true gradient with respect to $y$ accumulated over the epoch.

## B  Supporting Lemmas

In this section, we present several lemmas used to build our convergence analysis. Lemma B.1, Lemma B.2, Lemma B.3, and Lemma B.4 are fundamental properties of generalized smoothness provided by Li et al. (2024). We adapt them to the setting of hybrid system fine-tuning.

**Lemma B.1** (The generalized version of Lemma 3.3 from Li et al. (2024))**.** *Let $f : \mathbb{R}^d = \mathbb{R}^{d_x} \times \mathbb{R}^{d_y} \to \mathbb{R}$ be a twice continuously differentiable function satisfying the hybrid generalized smoothness properties. Suppose that $(x, y) \in \mathbb{R}^d$ satisfies $\|\nabla f(x, y)\| \leq G$. Then there exist non-negative constant $L_x = \ell_x(G)$ and $L_y = \ell_y(G)$ such that for all $(x_1, y_1), (x_2, y_2) \in \mathcal{B}(x, \frac{G}{L_x}) \times \mathcal{B}(y, \frac{G}{L_y})$:*

*1. $\|\nabla_x f(x_1, y') - \nabla_x f(x_2, y')\| \leq L_x \|x_1 - x_2\|$, for all $y' \in \mathbb{R}^{d_y}$.*

*2. $\|\nabla_y f(x', y_1) - \nabla_y f(x', y_2)\| \leq L_y \|y_1 - y_2\|$, for all $x' \in \mathbb{R}^{d_x}$.*

*3. Let $I_d$ represent the identity matrix with the size $d \times d$.*

$$f(x_1, y_1) \leq f(x_2, y_2) + \left\langle \nabla f(x_2, y_2), \begin{bmatrix} x_1 - x_2 \\ y_1 - y_2 \end{bmatrix} \right\rangle$$
$$+ \frac{1}{2} \begin{bmatrix} x_1 - x_2 & y_1 - y_2 \end{bmatrix} \begin{bmatrix} L_x I_{d_x} & 0 \\ 0 & L_y I_{d_y} \end{bmatrix} \begin{bmatrix} x_1 - x_2 \\ y_1 - y_2 \end{bmatrix}.$$

*Proof.* Let $(x, y) \in \mathbb{R}^d = \mathbb{R}^{d_x} \times \mathbb{R}^{d_y}$ be arbitrary. By the assumption of twice continuous differentiability and the mean value theorem, we have

$$\nabla_x f(x_2, y) - \nabla_x f(x_1, y) = \int_0^1 \nabla_{xx}^2 f(x_1 + t(x_2 - x_1), y)(x_2 - x_1) dt.$$

Taking the norm of both sides and applying the generalized smoothness of $f$ (Definition 2.1), we obtain

$$\|\nabla_{xx}^2 f(x, y)\| \leq \ell_x(\|\nabla f(x, y)\|) \leq L_x,$$

where the last inequality is by the monotonicity of $\ell_x$ and the bounded gradient condition. We apply this inequality to the integral yields the first inequality. The second inequality for the y-gradient is obtained similarly. For the third inequality, we still consider the mean value theorem:

$$\begin{aligned} f(x_1, y_1) - f(x_2, y_2) &= \int_0^1 \left\langle \nabla f(z(t), \begin{bmatrix} x_1 - x_2 \\ y_1 - y_2 \end{bmatrix} \right\rangle dt \\ &= \int_0^1 \left[ \left\langle \nabla f(x_2, y_2), \begin{bmatrix} x_1 - x_2 \\ y_1 - y_2 \end{bmatrix} \right\rangle + \left\langle \nabla f(z(t)) - \nabla f(x_2, y_2), \begin{bmatrix} x_1 - x_2 \\ y_1 - y_2 \end{bmatrix} \right\rangle \right] dt \\ &= \left\langle \nabla f(x_2, y_2), \begin{bmatrix} x_1 - x_2 \\ y_1 - y_2 \end{bmatrix} \right\rangle + \int_0^1 \left\langle \nabla f(z(t)) - \nabla f(x_2, y_2), \begin{bmatrix} x_1 - x_2 \\ y_1 - y_2 \end{bmatrix} \right\rangle dt \\ &\leq \left\langle \nabla f(x_2, y_2), \begin{bmatrix} x_1 - x_2 \\ y_1 - y_2 \end{bmatrix} \right\rangle + L_y \|y_1 - y_2\|^2 \int t dt + L_x \|x_1 - x_2\|^2 \int t dt, \end{aligned}$$

where $z(t) := (1 - t) \begin{bmatrix} x_2 \\ y_2 \end{bmatrix} + t \begin{bmatrix} x_1 \\ y_1 \end{bmatrix}$ for $0 \leq t \leq 1$. Then the proof is completed by re-arranging this inequality. $\square$

**Lemma B.2** (The generalized version of Lemma 3.5 from Li et al. (2024))**.** *Let $f : \mathbb{R}^{d_x} \times \mathbb{R}^{d_y} \to \mathbb{R}$ be a twice continuously differentiable function satisfying the hybrid generalized smoothness properties. Let $f^* = \inf_{x,y} f(x, y)$ be the global minimum of $f$. Then, for all $(x, y) \in \mathbb{R}^{d_x} \times \mathbb{R}^{d_y}$, the following inequalities hold:*

*1. $\|\nabla_x f(x, y)\|^2 \leq 2\ell_x(2\|\nabla f(x, y)\|) \cdot (f(x, y) - f^*)$*

*2. $\|\nabla_y f(x, y)\|^2 \leq 2\ell_y(2\|\nabla f(x, y)\|) \cdot (f(x, y) - f^*)$*

*3. $\frac{1}{2}[\nabla f(x, y)]^\top \begin{bmatrix} \frac{I_{d_x}}{\ell_x(2\|\nabla f(x,y)\|)} & 0 \\ 0 & \frac{I_{d_y}}{\ell_y(2\|\nabla f(x,y)\|)} \end{bmatrix} \nabla f(x, y) \leq f(x, y) - f^*.$*

*Proof.* The first and the second inequalities are directly implied by Lemma 3.5 from Li et al. (2024) by projecting the objective function $f$ to a subspace of the domain. Here, we provide the proof

for the third inequality. By Lemma B.1 where we choose $G = \|\nabla f(x,y)\|$, we have that for any $(x_1, y_1), (x_2, y_2) \in \mathcal{B}(x, \frac{G}{L_x}) \times \mathcal{B}(y, \frac{G}{L_y})$,

$$f(x_1, y_1) \leq f(x_2, y_2) + \left\langle \nabla f(x_2, y_2), \begin{bmatrix} x_1 - x_2 \\ y_1 - y_2 \end{bmatrix} \right\rangle + \frac{1}{2} \begin{bmatrix} x_1 - x_2 & y_1 - y_2 \end{bmatrix} \begin{bmatrix} L_x I_{d_x} & 0 \\ 0 & L_y I_{d_y} \end{bmatrix} \begin{bmatrix} x_1 - x_2 \\ y_1 - y_2 \end{bmatrix}.$$

Choosing $(x_2, y_2) = (x, y)$, $x_1 = x - \frac{\nabla_x f(x,y)}{\ell_x(2\|\nabla f(x,y)\|)}$, and $y_1 = y - \frac{\nabla_y f(x,y)}{\ell_y(2\|\nabla f(x,y)\|)}$, we obtain

$$f^* \leq f(x - \frac{\nabla_x f(x,y)}{\ell_x(2\|\nabla f(x,y)\|)}, y - \frac{\nabla_y f(x,y)}{\ell_y(2\|\nabla f(x,y)\|)})$$

$$\leq f(x,y) - \frac{1}{2}[\nabla f(x,y)]^\top \begin{bmatrix} \frac{I_{d_x}}{\ell_x(2\|\nabla f(x,y)\|)} & 0 \\ 0 & \frac{I_{d_y}}{\ell_y(2\|\nabla f(x,y)\|)} \end{bmatrix} \nabla f(x,y).$$

Then the proof is completed. $\qquad\qquad\square$

**Lemma B.3** (The generalized version of Corollary 3.6 from Li et al. (2024))**.** *Let $f : \mathbb{R}^{d_x} \times \mathbb{R}^{d_y} \to \mathbb{R}$ be a twice continuously differentiable function satisfying the hybrid generalized smoothness properties. Suppose that $f(x,y) - f^* \leq F$ for some $(x,y) \in \mathbb{R}^d$ and $F \geq 0$. Denoting $G := \sup\{u \geq 0 \mid u^2 \leq 2\max(\ell_x, \ell_y)(u) \cdot F\}$, then $\|\nabla f(x,y)\| \leq G < \infty$.*

*Proof.* Let $\max(\ell_x, \ell_y)(u) := \max\{\ell_x(u), \ell_y(u)\}$. Since both $\ell_x$ and $\ell_y$ are sub-quadratic, it concludes $G$ is finite (by Corollary 3.6 from Li et al. (2024)). From Lemma B.2, we have

$$\frac{1}{2}[\nabla f(x,y)]^\top \begin{bmatrix} \frac{I_{d_x}}{\max(\ell_x, \ell_y)(2\|\nabla f(x,y)\|)} & 0 \\ 0 & \frac{I_{d_y}}{\max(\ell_x, \ell_y)(2\|\nabla f(x,y)\|)} \end{bmatrix} \nabla f(x,y)$$

$$\leq \frac{1}{2}[\nabla f(x,y)]^\top \begin{bmatrix} \frac{I_{d_x}}{\ell_x(2\|\nabla f(x,y)\|)} & 0 \\ 0 & \frac{I_{d_y}}{\ell_y(2\|\nabla f(x,y)\|)} \end{bmatrix} \nabla f(x,y)$$

$$\leq f(x,y) - f^*.$$

Therefore, we obtain

$$\|\nabla f(x,y)\|^2 \leq 2\max(\ell_x, \ell_y)(2\nabla f(x,y)) \cdot F.$$

It concludes that if the function value is bounded, then the gradient is also bounded. $\qquad\square$

Here, we summarize the previous results in the following lemma. The constant $G$ (determined by the function value upper bound $F$) is defined in Lemma B.3 and the constant $L_x$ and $L_y$ (determined by the gradient norm upper bound $G$) is defined in Lemma B.1.

**Lemma B.4.** *Suppose that Assumption 3.2 holds for the objective function $f(x,y) := \frac{1}{n}\sum_{i=1}^n f(x,y;i)$, with all individual loss functions $f(\cdot;i)$ are twice continuously differentiable and satisfy the hybrid generalized smoothness properties. Let $\mathcal{G}_F := \{(x,y) \in \mathbb{R}^d \mid f(x,y) - f^* \leq F\}$. Then the following statements hold:*

1. *The objective function $f(\cdot)$ has $G$-bounded gradient over $\mathcal{G}_F$; that is, $\|\nabla f(x,y)\| \leq G$ for all $(x,y) \in \mathcal{G}_F$.*

2. *The objective function $f(\cdot)$ has $(L_x, L_y)$-Lipschitz gradient over $\mathcal{G}_F$; that is, $\|\nabla_x f(x,y) - \nabla_x f(x', y)\| \leq L_x\|x - x'\|$ and $\|\nabla_y f(x,y) - \nabla_y f(x,y')\| \leq L_y\|y - y'\|$ for all $(x,y), (x', y') \in \mathcal{G}_F$.*

3. *The individual loss function $f(\cdot;i)$ has $(G_{x,\max}, G_{y,\max})$-bounded gradient over $\mathcal{G}_F$; that is, $\|\nabla_x f(x,y;\xi)\| \leq G_{x,\max}$ and $\|\nabla_y f(x,y;\xi)\| \leq G_{y,\max}$ for all $(x,y) \in \mathcal{G}_F$ and all $\xi \in \{1, 2, \ldots, n\}$.*

4. *The individual loss function $f(\cdot;i)$ has $(L_{x,\max}, L_{y,\max})$-Lipschitz gradient over $\mathcal{G}_F$; that is, $\|\nabla_x f(x,y;\xi) - \nabla_x f(x',y;\xi)\| \leq L_{x,\max}\|x - x'\|$ and $\|\nabla_y f(x,y;\xi) - \nabla_y f(x,y';\xi)\| \leq L_{y,\max}\|y - y'\|$ for all $(x,y) \in \mathcal{G}_F$ and all $\xi \in \{1, 2, \ldots, n\}$.*

*Proof.* By Assumption 3.2, $\mathcal{G}_F$ is a compact set. By the twice continuous differentiability of the objective function $f(\cdot)$ (and all individual loss functions $f(\cdot;i)$), all statements holds by its continuity. More precise evaluation is given in Lemma B.1 for $L_x$ and $L_y$, and in Lemma B.3 for $G$. $\qquad\square$

The following lemma characterizes the accuracy of zeroth-order gradient estimation. We note that the choice of zeroth-order gradient estimator is not the crucial part in our analysis; the following gradient estimation method can be replaced with any common zeroth-order optimization techniques, including the mini-batch zeroth-order gradient estimation (Nesterov & Spokoiny, 2017), the uniform smoothing (Gasnikov et al., 2022), and the variance reduction (Liu et al., 2018).

**Lemma B.5.** *Let $f : \mathbb{R}^d \to \mathbb{R}$ be a function with twice continuous differentiability. Define the two-point zeroth-order gradient estimator of $\nabla f(x)$ as*

$$\hat{\nabla} f(x) := \frac{1}{\mu} \left[ f(x + \mu v) - f(x) \right] v,$$

*where $\mu > 0$ is the perturbation stepsize, $v \in \mathbb{R}^d$ is a Gaussian vector with the covariance matrix $I_d$. Suppose that $f$ has $G$-bounded gradient and $L$-Lipschitz gradient at $x$. Then*

*1.* $\mathbf{E}\langle g, \hat{\nabla} f(x) - \nabla f(x)\rangle \leq \frac{\mu}{2} L(d+3)^{3/2} \|g\|$, *for any* $g \in \mathbb{R}^d$.

*2.* $\mathbf{E}\|\hat{\nabla} f(x) - \nabla f(x)\|^2 \leq 32d\|\nabla f(x)\|^2 + 108\mu^2 L^2 d^4$.

*Proof.* Throughout this proof, we follow the *random gradient-free oracles* given by Nesterov & Spokoiny (2017). That is, define

$$f_\mu(x) = \mathbf{E}_{v \sim N(0, I_d)} f(x + \mu v);$$

then the gradient estimator $\hat{\nabla} f(x)$ is an unbiased estimator of $\nabla f_\mu(x)$. For the first inequality, we have

$$\mathbf{E}\langle g, \hat{\nabla} f(x) - \nabla f(x)\rangle \overset{(i)}{=} \mathbf{E}\langle g, \nabla f_\mu(x) - \nabla f(x)\rangle$$
$$\overset{(ii)}{=} \frac{\mu}{2} L(d+3)^{3/2} \|g\|.$$

where (i) applies the unbiasedness of Gaussian smoothing and (ii) applies Lemma 3 from Nesterov & Spokoiny (2017). For the second inequality, we have

$$\mathbf{E}\|\hat{\nabla} f(x) - \nabla f(x)\|^2 \leq 2\mathbf{E}\|\hat{\nabla} f(x)\|^2 + 2\|\nabla f(x)\|^2$$
$$\overset{(i)}{\leq} 8(d+4)\|\nabla f_\mu(x)\|^2 + 6\mu^2 L^2 (d+4)^3 + 2\|\nabla f(x)\|^2$$
$$\overset{(ii)}{\leq} 32d\|\nabla f(x)\|^2 + 108\mu^2 L^2 d^4.$$

where (i) applies Lemma 5 from Nesterov & Spokoiny (2017) and (ii) again applies Lemma 3 from Nesterov & Spokoiny (2017). $\qquad\square$

**Lemma B.6.** *Suppose that Assumption 3.2 and Assumption 3.3 hold for the objective function $f(x, y) := \frac{1}{n} \sum_{i=1}^{n} f(x, y; i)$, with all individual loss functions $f(\cdot; i)$ are twice continuously differentiable and satisfy the hybrid generalized smoothness properties. Let*

$$\epsilon_t = \frac{1}{n} \sum_{i=1}^{n} \hat{\nabla} f(x_{t,i}, y_{t,i}; \xi_{t,i}) - \frac{1}{n} \sum_{i=1}^{n} \nabla f(x_{t,i}, y_{t,i}; \xi_{t,i}) + \frac{1}{n} \sum_{i=1}^{n} \nabla f(x_{t,i}, y_{t,i}; \xi_{t,i}) - \nabla f(x_t, y_t),$$

*be the gradient approximation error over the $t$-th epoch. Given any $F, H > 0$, define the stopping time as $\tau = \tau_1 \wedge \tau_2$, where $\tau_1 := \min_t \{t \mid f(x_{t+1}, y_{t+1}) - f^* > F\} \wedge T$ and $\tau_2 := \min_t \{t \mid \|\epsilon_t\| > H\} \wedge T$. Let the learning rates satisfy $\eta_x \leq \min\{\frac{1}{2L_{x,\max}n}, \frac{1}{384L_x n d_x}\}$ and $\eta_y \leq \frac{1}{2L_{y,\max}n}$ and the perturbation stepsize $\mu \leq \frac{G}{L_x} \frac{6}{d_x^{3/2}}$. Then*

$$f(x_\tau, y_\tau) - f^* + \sum_{t < \tau} [\nabla f(x_t, y_t)]^\top \begin{bmatrix} \frac{n}{4}\eta_x I_{d_x} & 0 \\ 0 & \frac{n}{3}\eta_y I_{d_y} \end{bmatrix} \nabla f(x_t, y_t)$$

$$\leq f_0 - f^* + \left[ \frac{\sigma^2}{2} n^2 \left[ \eta_y^3 L_{y,\max}^2 + \eta_x^3 L_{x,\max}^2 \right] + o(\mu) \right] T,$$

*where $o(\mu) \leq 3\eta_x \mu n L_x d_x G$ is a small error term when $\mu$ is chosen small.*

*Proof.* For arbitrary stopping time $\tau$, we start from the smoothness given by Lemma B.1:

$$f(x_{t+1}, y_{t+1}) - f(x_t, y_t)$$

$$\leq \left\langle \nabla f(x_t, y_t), \begin{bmatrix} x_{t+1} - x_t \\ y_{t+1} - y_t \end{bmatrix} \right\rangle + \frac{1}{2} [x_{t+1} - x_t \quad y_{t+1} - y_t] \begin{bmatrix} L_x I_{d_x} & 0 \\ 0 & L_y I_{d_y} \end{bmatrix} \begin{bmatrix} x_{t+1} - x_t \\ y_{t+1} - y_t \end{bmatrix}$$

$$= \langle \nabla_x f(x_t, y_t), x_{t+1} - x_t \rangle + \langle \nabla_y f(x_t, y_t), x_{t+1} - x_t \rangle + \frac{L_x}{2} \|x_{t+1} - x_t\|^2 + \frac{L_y}{2} \|y_{t+1} - y_t\|^2$$

$$\overset{(i)}{=} -\eta_x n \langle \nabla_x f(x_t, y_t), \frac{\hat{g}_t}{n} - \frac{g_t}{n} \rangle - \eta_x n \langle \nabla_x f(x_t, y_t), \frac{g_t}{n} \rangle + \eta_x^2 L_x n^2 \|\frac{\hat{g}_t}{n} - \frac{g_t}{n}\|^2 + \eta_x^2 L_x n^2 \|\frac{g_t}{n}\|^2$$

$$- \eta_y n \langle \nabla_y f(x_t, y_t), \frac{h_t}{n} \rangle + \eta_y^2 L_y n^2 \|\frac{h_t}{n}\|^2,$$

where (i) we applies the derivation of Eq.(38) from Mishchenko et al. (2020) with setting $\eta_x \leq \frac{1}{2L_x}$ and $\eta_y \leq \frac{1}{2L_y}$. We note that the y parameter update doesn't involve the gradient estimation; so, we keep the original stochastic gradient $h_t$ for this step. Let $\mathcal{E}_1 = -\eta_x n \langle \nabla_x f(x_t, y_t), \frac{\hat{g}_t}{n} - \frac{g_t}{n} \rangle$ and $\mathcal{E}_2 = \eta_x^2 L_x n^2 \|\frac{\hat{g}_t}{n} - \frac{g_t}{n}\|^2$, representing the errors caused by the zeroth-order gradient estimation. Then we obtain

$$f(x_{t+1}, y_{t+1}) - f(x_t, y_t) \leq -\eta_x n \langle \nabla_x f(x_t, y_t), \frac{g_t}{n} \rangle + \eta^2 L_x n^2 \|\frac{g_t}{n}\|^2 + \mathcal{E}_1 + \mathcal{E}_2$$

$$- \eta_y n \langle \nabla_x f(x_t, y_t), \frac{h_t}{n} \rangle + \eta^2 L_y n^2 \|\frac{h_t}{n}\|^2.$$

Then we set $\eta_x \leq \frac{1}{2L_x n}$ and $\eta_y \leq \frac{1}{2L_y n}$. By Eq.(39) from Mishchenko et al. (2020),

$$f(x_{t+1}, y_{t+1}) - f(x_t, y_t) + \frac{\eta_x n}{2} \|\nabla_x f(x_t, y_t)\|^2 + \frac{\eta_y n}{2} \|\nabla_y f(x_t, y_t)\|^2$$

$$\leq \frac{\eta_x n}{2} \left\| \frac{g_t}{n} - \nabla_x f(x_t, y_t) \right\|^2 + \frac{\eta_y n}{2} \left\| \frac{h_t}{n} - \nabla_y f(x_t, y_t) \right\|^2 + \mathcal{E}_1 + \mathcal{E}_2.$$

Then we take expectation on both sides and decompose $\left\| \nabla_x f(x_t, y_t) - \frac{g_t}{n} \right\|^2$ using Lemma B.1 with the Lipschitz constant $L_{x,\max}$ and $\left\| \nabla_y f(x_t, y_t) - \frac{g_t}{n} \right\|^2$ with the Lipschitz constant $L_{y,\max}$; more explicitly, we have

$$\left\| \nabla_x f(x_t, y_t) - \frac{g_t}{n} \right\|^2 = \left\| \frac{1}{n} \sum_{i=1}^n \nabla_x f(x_{t,0}, y_{t,0}; \xi_{t,i}) - \frac{1}{n} \sum_{i=1}^n \nabla_x f(x_{t,i}, y_{t,i}; \xi_{t,i}) \right\|^2$$

$$\leq \frac{1}{n} \sum_{i=1}^n \|\nabla_x f(x_{t,0}, y_{t,0}; \xi_{t,i}) - \nabla_x f(x_{t,i}, y_{t,i}; \xi_{t,i})\|^2$$

$$\leq \frac{L_{x,\max}^2}{n} \sum_{i=1}^n \|x_{t,0} - x_{t,i}\|^2.$$

Applying Assumption 3.3 and Lemma 5 from Mishchenko et al. (2020) to bound $\frac{L_{x,\max}^2}{n} \sum_{i=1}^n \mathbf{E} \|x_{t,0} - x_{t,i}\|^2$, we obtain

$$f(x_{t+1}, y_{t+1}) - f(x_t, y_t) + \frac{\eta_x n}{2} \|\nabla_x f(x_t, y_t)\|^2 + \frac{\eta_y n}{2} \|\nabla_y f(x_t, y_t)\|^2$$

$$\leq \frac{\eta_x n}{2} \frac{L_{x,\max}^2}{n} [\eta_x^2 n^3 \|\nabla_x f(x_t, y_t)\|^2 + \eta_x^2 n^2 \sigma^2] + \frac{\eta_y n}{2} \frac{L_{y,\max}^2}{n} [\eta_y^2 n^3 \|\nabla f(x_t, y_t)\|^2 + \eta_y^2 n^2 \sigma^2] + \mathbf{E}\mathcal{E}_1 + \mathbf{E}\mathcal{E}_2.$$

We re-write this inequality into the matrix form.

$$f(x_{t+1}, y_{t+1}) - f(x_t, y_t) + [\nabla f(x_t, y_t)]^\top \begin{bmatrix} \frac{\eta_x n}{2} & 0 \\ 0 & \frac{\eta_y n}{2} \end{bmatrix} \nabla f(x_t, y_t)$$

$$\leq \frac{\sigma^2}{2} n^2 [\eta_x^3 L_{x,\max}^2 + \eta_y^3 L_{y,\max}^2] + \mathbf{E}\mathcal{E}_1 + \mathbf{E}\mathcal{E}_2 + [\nabla f(x_t, y_t)]^\top \begin{bmatrix} \frac{\eta_x^3 n^3 L_{x,\max}^2}{2} I_{d_x} & 0 \\ 0 & \frac{\eta_y^3 n^3 L_{y,\max}^2}{2} I_{d_y} \end{bmatrix} \nabla f(x_t, y_t).$$

When choosing $\eta_x \leq \frac{1}{2L_{x,\max}n}$ and $\eta_y \leq \frac{1}{2L_{y,\max}n}$, it ensures that

$$\frac{n}{3}\begin{bmatrix} \eta_x I_{d_x} & 0 \\ 0 & \eta_y I_{d_x} \end{bmatrix} \preceq \begin{bmatrix} \frac{\eta_x n}{2}I_{d_x} & 0 \\ 0 & \frac{\eta_y n}{2}I_{d_y} \end{bmatrix} - \begin{bmatrix} \frac{\eta_x^3 n^3 L_{x,\max}^2}{2}I_{d_x} & 0 \\ 0 & \frac{\eta_y^3 n^3 L_{y,\max}^2}{2}I_{d_y} \end{bmatrix}.$$

Therefore, we let $\Lambda^2 = \frac{n}{3}\begin{bmatrix} \eta_x I_{d_x} & 0 \\ 0 & \eta_y I_{d_y} \end{bmatrix}$ be a PSD matrix. Then we obtain

$$f(x_{t+1}, y_{t+1}) - f(x_t, y_t) + \|\Lambda \nabla f(x_t, y_t)\|^2 \leq \frac{\sigma^2}{2}n^2\left[\eta_x^3 L_{x,\max}^2 + \eta_y^3 L_{y,\max}^2\right] + \mathbf{E}\mathcal{E}_1 + \mathbf{E}\mathcal{E}_2.$$

Then we apply [Lemma B.5](#) to bound $\mathbf{E}\mathcal{E}_1$ and $\mathbf{E}\mathcal{E}_2$, respectively. By the stopping time construction, we have $\|\nabla_x f(x_t, y_t)\| \leq \|\nabla f(x_t, y_t)\| \leq G$. Therefore, we have

$$\mathbf{E}\mathcal{E}_1 = -\eta_x n \mathbf{E}\langle \nabla_x f(x_t, y_t), \frac{\hat{g}_t}{n} - \frac{g_t}{n}\rangle$$
$$\leq \eta_x \frac{\mu n}{2}L_x(d_x + 3)^{3/2}G.$$

Similarly, we have

$$\mathbf{E}\mathcal{E}_2 = \eta_x^2 L_x n^2 \mathbf{E}\|\frac{\hat{g}_t}{n} - \frac{g_t}{n}\|^2$$
$$\leq \eta_x^2 L_x n^2 \left[32d_x\|\nabla_x f(x_t, y_t)\|^2 + 108\mu^2 L^2 d^4\right].$$

We further simply the inequality by letting $\eta_x \leq \frac{1}{384L_x nd}$. Then we have

$$f(x_{t+1}, y_{t+1}) - f(x_t, y_t) + [\nabla f(x_t, y_t)]^\top \begin{bmatrix} \frac{n}{4}\eta_x I_{d_x} & 0 \\ 0 & \frac{n}{3}\eta_y I_{d_y} \end{bmatrix} \nabla f(x_t, y_t)$$
$$\leq \frac{\sigma^2}{2}n^2\left[\eta_y^3 L_{y,\max}^2 + \eta_x^3 L_{x,\max}^2\right] + o(\mu),$$

where $o(\mu)$ represents a small error term when $\mu$ tends to 0. Lastly, we sum over $t < \tau$ and obtain

$$f(x_\tau, y_\tau) - f^* + \sum_{t<\tau}[\nabla f(x_t, y_t)]^\top \begin{bmatrix} \frac{n}{4}\eta_x I_{d_x} & 0 \\ 0 & \frac{n}{3}\eta_y I_{d_y} \end{bmatrix} \nabla f(x_t, y_t)$$
$$\leq f_0 - f^* + \left[\frac{\sigma^2}{2}n^2\left[\eta_y^3 L_{y,\max}^2 + \eta_x^3 L_{x,\max}^2\right] + o(\mu)\right]T,$$

which completes the proof. Here, $o(\mu) \leq 3\eta_x \mu n L_x dG$ by letting $\mu \leq \frac{G}{L_x}\frac{6}{d_x^{3/2}}$. $\qquad\square$

## C    PROOF OF [THEOREM 3.4](#)

Here, we re-state our main theorem with full details.

**Theorem C.1.** *Suppose that [Assumption 3.2](#) and [Assumption 3.3](#) hold for the objective function $f(x, y) := \frac{1}{n}\sum_{i=1}^{n} f(x, y; i)$ and satisfy the hybrid generalized smoothness properties. Let $\delta \in (0, 1)$ and $\{(x_t, y_t)\}_{t=1}^{T}$ be the SGD with Random Shuffling dynamic generated by [Algorithm 1](#) for solving the optimization problem [Eq. (1)](#). Given $F$ as*

$$F = \frac{8}{\delta}[f_0 - f^* + \sigma'],$$

*where $f_0 := f(x_0, y_0)$ is the initial function value and $\sigma'$ is a constant-level value given by [Eq. (4)](#) and $H$ as*

$$H = 2\sqrt{\frac{[200G^2\frac{d_x}{n} + G^2 + \frac{\sigma^2}{n}]T}{\delta}},$$

*define the stopping time as $\tau = \tau_1 \wedge \tau_2$, where $\tau_1 := \min_t\{t \mid f(x_{t+1}, y_{t+1}) - f^* > F\} \wedge T$ and $\tau_2 := \min_t\{t \mid \|\epsilon_t\| > H\} \wedge T$, where $\epsilon_t$ is defined in Lemma B.6. If learning rates $\eta_x$, $\eta_y$, and the perturbation stepsize $\mu$ are chosen such that*

$$\eta_x \leq \min\left\{\frac{1}{2L_{x,\max}n}, \frac{1}{384L_x nd}, \sqrt{\frac{2}{T}}\frac{1}{\sigma n L_{x,\max}}\right\},$$

$$\eta_y \leq \min\left\{\frac{1}{2L_{y,\max}n}, \sqrt{\frac{2}{T}}\frac{1}{\sigma n L_{y,\max}}\right\}, \tag{3}$$

$$\mu \leq \min\left\{\frac{G}{L_x}\frac{6}{d^{3/2}}, \frac{1}{3L_x TndG}\right\}.$$

*where all constant $G, L_{x,\max}, L_{y,\max}, L_x, L_y$ are defined relying on $F$ with presented in Lemma B.4, and the maximum number of epoch $T$ is chosen as*

$$T \geq \epsilon^{-2}\left[\frac{2}{\delta} + \frac{G^2}{8}\right] + \epsilon^{-4}\left[\frac{f_0 - f^* + 3}{n}\right],$$

*then with the probability at least $1 - \delta$,*

$$\frac{1}{T}\sum_{t<T}\mathbf{E}\left\|\nabla f\left(x_t, y_t\right)\right\|^2 \leq \epsilon^2.$$

*Proof.* Let $A := \left\{\frac{1}{T}\sum_{t<T}\|\nabla f\left(x_t, y_t\right)\|^2 \leq \epsilon^2\right\}$ and $B := \{\tau \geq T\}$ be two events. We consider the following lower bound of the probability of event $A$ by conditioning it on the event $B$:

$$\mathbb{P}(A) \geq \mathbb{P}(A \cap B) = \mathbb{P}(A|B)\mathbb{P}(B)$$
$$\geq [1 - \mathbb{P}(A^c|B)][1 - \mathbb{P}(B^c)].$$

Our goal is to show that the probability of $\left\{\frac{1}{T}\sum_{t<T}\|\nabla f\left(x_t, y_t\right)\|^2 > \epsilon^2 \Big| \tau \geq T\right\}$ (the event $A^c|B$) and $\{\tau < T\}$ (the event $B^c$) are both small. We bound each term separately.

- First, we bound the probability of $\left\{\frac{1}{T}\sum_{t<T}\|\nabla f\left(x_t, y_t\right)\|^2 > \epsilon^2 \Big| \tau \geq T\right\}$. By Lemma B.6, we let

$$\sigma' = \left[\frac{\sigma^2}{2}n^2\left[\eta_y^3 L_{y,\max}^2 + \eta_x^3 L_{x,\max}^2\right] + o(\mu)\right]T. \tag{4}$$

If the event is conditioned on $\tau \geq T$, we always have $\|\nabla f\left(x_t\right)\| \leq G$ for $t = 1, 2, \ldots, T - 1$, where $G$ is determined by Lemma B.3. Then we obtain

$$\mathbb{P}\left(\sum_{t<T}\|\nabla f\left(x_t, y_t\right)\|^2 > c\Big|\tau \geq T\right) \overset{(i)}{\leq} \mathbb{P}\left(e^{\sum_{t<T}\|\nabla f(x_t, y_t)\|^2} > e^c\Big|\tau \geq T\right)$$

$$\overset{(ii)}{\leq} \mathbf{E}\left[e^{\sum_{t<T}\|\nabla f(x_t, y_t)\|^2}\Big|\tau \geq T\right]/e^c$$

$$\overset{(iii)}{\leq} \exp\left(\sum_{t<T}\mathbf{E}\|\nabla f\left(x_t\right)\|^2 + \frac{G^2}{8}\right)/e^c$$

$$\overset{(iv)}{\leq} \exp\left(\frac{1}{\eta_{\min}n}[f_0 - f^* + \sigma'] + \frac{G^2}{8} - c\right).$$

where (i) takes exponential on both sides, (ii) applies the Markov inequality, (iii) applies the Hoeffding's lemma, (iv) applies Lemma B.6 with setting $\eta_{\min} = \min\{\frac{\eta_x}{4}, \frac{\eta_y}{3}\}$ and $f_0 := f(x_0, y_0)$.

Before we evaluate the necessary $T$, we need to choose hyper-parameters to make $\sigma'$ less than some constant independent of $d$, $n$, or other crucial constants. To do so, we set

$$\eta_x \leq \sqrt{\frac{2}{T}}\frac{1}{\sigma n L_{x,\max}}, \quad \eta_y \leq \sqrt{\frac{2}{T}}\frac{1}{\sigma n L_{y,\max}}, \quad \mu \leq \frac{1}{3L_x Tnd_x G}.$$

Then we obtain $\sigma' \leq 2\eta_x + \eta_y$. Let $c = T\epsilon^2$ and $e^{\frac{1}{\eta_{\min}n}[f_0 - f^* + 2\eta_x + \eta_y] + \frac{G^2}{8}} e^{-c} \leq \frac{\delta}{2}$. Then it solves

$$\epsilon^2 T \geq \ln(\frac{2}{\delta}) + \frac{G^2}{8} + \frac{1}{\eta_{\min}n}[f_0 - f^* + 2\eta_x + \eta_y]$$

$$T \geq \epsilon^{-2}\left[\frac{2}{\delta} + \frac{G^2}{8}\right] + \epsilon^{-2}\left[\frac{f_0 - f^* + 2\eta_x + \eta_y}{\eta_{\min}n}\right].$$

- Then we bound the probability $\mathbb{P}(B^c) = \mathbb{P}(\tau < T)$. Recap that we consider the stopping time defined as $\tau = \tau_1 \wedge \tau_2$, where $\tau_1 := \min_t\{t \mid f(x_{t+1}, y_{t+1}) - f^* > F\} \wedge T$ and $\tau_2 := \min_t\{t \mid \|\epsilon_t\| > H\} \wedge T$. Here, $\epsilon_t$ is defined as

$$\epsilon_t = \underbrace{\frac{1}{n}\sum_{i=1}^n \hat{\nabla}f(x_{t,i};\xi_{t,i}) - \frac{1}{n}\sum_{i=1}^n \nabla f(x_{t,i};\xi_{t,i})}_{\text{est. err.}} + \underbrace{\frac{1}{n}\sum_{i=1}^n \nabla f(x_{t,i};\xi_{t,i}) - \nabla f(x_t)}_{\text{stoc. err.}}. \quad (5)$$

We note that for the last $d_y$ entries, the estimation error term is $0$ since we do not apply gradient estimation for this part. Both $F$ and $H$ in the definition of stopping times will be determined later. Then we notice that

$$\mathbb{P}(B^c) = \mathbb{P}(\tau < T) = \mathbb{P}(\{\tau_1 < T\} \cup \{\tau_2 < T\})$$
$$= \mathbb{P}(\tau_2 < T) + \mathbb{P}(\tau_1 < T, \tau_2 \geq T).$$

We bound each term separately as follows:

○ Choose $H$ such that $\mathbb{P}(\tau_2 < T) \leq \frac{\delta}{4}$: We have

$$\mathbb{P}(\tau_2 < T) = \mathbb{P}\left(\bigcup_{t<T}\{\|\epsilon_t\| > H\}\right)$$

$$\leq \sum_{t<T}\mathbb{P}\left(\|\epsilon_t\| > H\right)$$

$$\overset{(i)}{\leq} \sum_{t<T}\frac{\frac{3}{n^2}\mathbf{E}\|g_t - \hat{g}_t\|^2 + 3\mathbf{E}\|\frac{g_t}{n} - \nabla_x f(x_t, y_t)\|^2 + 3\mathbf{E}\|\frac{h_t}{n} - \nabla_y f(x_t, y_t)\|^2}{H^2}$$

$$\overset{(ii)}{\leq} \left[\frac{3}{n}\left[64d\|\nabla_x f(x_t, y_t)\|^2 + 216\mu^2 L_{x,\max}^2 d_x^4\right]/H^2\right.$$

$$\left. + \left(3L_{x,\max}^2\eta_x^2 + 3L_{y,\max}^2\eta_y^2\right)\left[n^2 G^2 + n\sigma^2\right]/H^2\right]T$$

$$\overset{(iii)}{\leq} \frac{\left[200G^2\frac{d_x}{n} + 2G^2 + \frac{\sigma^2}{n}\right]T}{H^2}$$

where (i) applies the Markov inequality, (ii) applies Lemma B.5 and Lemma 5 from Mishchenko et al. (2020), and (iii) we choose a sufficiently small $\mu \leq \frac{8G}{L_{x,\max}d_x^{3/2}}$ and learning rates $\eta_x \leq \frac{1}{\sqrt{3}L_{x,\max}n}$ and $\eta_y \leq \frac{1}{\sqrt{3}L_{y,\max}n}$ to simplify the upper bound. Then we choose $\frac{\left[200G^2\frac{d_x}{n} + 2G^2 + \frac{\sigma^2}{n}\right]T}{H^2} = \frac{\delta}{4}$. It solves

$$H = 2\sqrt{\frac{[200G^2\frac{d_x}{n} + G^2 + \frac{\sigma^2}{n}]T}{\delta}}. \quad (6)$$

○ Choose $F$ such that $\mathbb{P}(\tau_1 < T, \tau_2 \geq T) \leq \frac{\delta}{4}$. Because $\{\tau_1 < T, \tau_2 \geq T\} \subset \{f(x_\tau, y_\tau) - f^* > \frac{F}{2}\}$,

$$\mathbb{P}(\tau_1 < T, \tau_2 \geq T) \leq \mathbb{P}(f(x_\tau, y_\tau) - f^* > \frac{F}{2})$$

$$\overset{(i)}{\leq} 2\mathbf{E}[f(x_\tau, y_\tau) - f^*]/F$$

$$\leq 2[f_0 - f^* + \sigma']/F.$$

where (i) applies the Markov inequality. Let $\frac{\delta}{4} = 2[f(x_0) - f^* + \sigma']/F$. It solves

$$F = \frac{8}{\delta}[f(x_0) - f^* + \sigma'].\tag{7}$$

Combining both upper bounds with choosing $H$ and $F$ defined by Eq. (6) and Eq. (7), respectively, we have

$$\mathbb{P}(B^c) = \mathbb{P}(\tau < T) \leq \frac{\delta}{2}.$$

Then we obtain the lower bound of $\mathbb{P}(A \cap B)$ as follows:

$$\mathbb{P}(A \cap B) = \mathbb{P}(A|B)\mathbb{P}(B) \geq [1 - \mathbb{P}(A^c|B)][1 - \mathbb{P}(B^c)]$$

$$\geq [1 - \frac{\delta}{2}][1 - \frac{\delta}{2}] = 1 - \delta + \frac{\delta^2}{4}$$

$$\geq 1 - \delta.$$

Lastly, we discuss the hyper-parameter choices and the epoch complexity. To make Lemma B.6 hold, we have set $\eta_x \leq \min\{\frac{1}{2L_{x,\max}n}, \frac{1}{384L_x nd_x}\}$ and $\eta_y \leq \frac{1}{2L_{y,\max}n}$ and the perturbation stepsize $\mu \leq \frac{G}{L_x}\frac{6}{d_x^{3/2}}$. When bounding the probability of $\left\{\frac{1}{T}\sum_{t<T}\|\nabla f(x_t, y_t)\|^2 > \epsilon^2 \middle| \tau \geq T\right\}$ and the probability of $\mathbb{P}(\tau < T)$, we additionally require

$$\eta_x \leq \min\{\sqrt{\frac{2}{T}}\frac{1}{\sigma n L_{x,\max}}, \frac{1}{\sqrt{3}L_{x,\max}n}\},$$

$$\eta_y \leq \min\{\sqrt{\frac{2}{T}}\frac{1}{\sigma n L_{y,\max}}, \frac{1}{\sqrt{3}L_{y,\max}n}\},$$

$$\mu \leq \min\{\frac{1}{3L_x Tnd_x G}, \frac{8G}{L_{x,\max}d_x^{3/2}}\}.$$

Therefore, in summary, we have

$$\eta_x \leq \min\left\{\frac{1}{2L_{x,\max}n}, \frac{1}{384L_x nd_x}, \sqrt{\frac{2}{T}}\frac{1}{\sigma n L_{x,\max}}\right\},$$

$$\eta_y \leq \min\left\{\frac{1}{2L_{y,\max}n}, \sqrt{\frac{2}{T}}\frac{1}{\sigma n L_{y,\max}}\right\},$$

$$\mu \leq \min\left\{\frac{G}{L_x}\frac{6}{d_x^{3/2}}, \frac{1}{3L_x Tnd_x G}\right\}.$$

Under these hyper-parameter choices, we also need to require

$$T \geq \epsilon^{-2}\left[\frac{2}{\delta} + \frac{G^2}{8}\right] + \epsilon^{-2}\left[\frac{f_0 - f^* + 2\eta_x + \eta_y}{\eta_{\min}n}\right],$$

where $\eta_{\min} = \min\{\frac{\eta_x}{4}, \frac{\eta_y}{3}\}$, to ensure that the probability of $\left\{\frac{1}{T}\sum_{t<T}\|\nabla f(x_t, y_t)\|^2 > \epsilon^2 \middle| \tau \geq T\right\}$ is small (less than $\frac{\delta}{2}$). We observe that by simply setting $\eta_{\min} \leq \epsilon^2$ (we can always make it by choosing $T \geq \Theta(\frac{\epsilon^{-4}}{n^2})$), the above condition on $T$ degenerates to $T \geq \Theta(\frac{\epsilon^{-4}}{n})$. Therefore, it concludes that if $T = \Theta(\epsilon^{-4}/n)$, with the probability at least $1 - \delta$,

$$\frac{1}{T}\sum_{t<T}\|\nabla f(x_t)\|^2 \leq \epsilon^2.$$

Then the proof is completed.

Here, we discuss how we determine the optimal value $\eta_{\min} = \Theta(\epsilon^2)$. In general, we can set $\eta_{\min} = \Theta(\epsilon^\alpha)$, which leads to the condition on $T$: $T \geq \Theta(\epsilon^{-2-\alpha})$. A smaller $\alpha$ is always better. However, we need to ensure the learning rate condition is satisfied; that is, $\eta_{\min} \leq \Theta(\sqrt{\frac{1}{T}})$. It solves $T \leq \Theta(\epsilon^{-2\alpha})$. We let $\epsilon^{-2\alpha} \geq \epsilon^{-2-\alpha}$, which solves $\alpha \geq 2$. Therefore, when $\eta_{\min} = \Theta(\epsilon^2)$, the complexity is optimal and attainable. $\qquad\square$

## D    EXPERIMENTS DETAILS

In this paper, we evaluate our proposed hybrid-tuning method across a diverse spectrum of scenarios including three distinct tasks, two transformer-based language models, and three PEFT methods. This extensive exploration not only demonstrates the broad applicability of our approach but also provides robust evidence for its effectiveness and versatility in enhancing model performance across various domains and architectures. In this section, we will briefly review these components and delve into more details of our experiment settings.

### D.1    OVERVIEW OF TASKS

In this section, we briefly discuss the task we consider in our paper. All of tasks are ready to use in the ZO-Bench code base (Zhang et al., 2024) and we follow the default setting and the same train/test/validation split of their original implementations.

**Text Binary Classification**    In this paper, we consider the Stanford Sentiment Treebank v2 (SST2) dataset (Socher et al., 2013) and the Word-In-Context (WIC) dataset (Pilehvar & Camacho-Collados, 2018), which presents the simplest binary text classification problem. The SST2 dataset is sufficiently simple and convenience to use to verify our motivating examples (as demonstrated in Figure 1a and Figure 1b). The WIC dataset provides a more challenging task that requires understanding word meanings in different contexts. Both datasets serve as excellent benchmarks for evaluating the performance of our proposed methods in binary text classification tasks.

**Question Answering**    The Choice Of Plausible Alternatives (COPA) dataset (Roemmele et al., 2011) is a common benchmark for evaluating the commonsense causal reasoning ability of a language model. It contains one thousand English-language questions answer pairs. We choose this task to evaluate our approaches in improving the question-answering capabilities of models, particularly in scenarios requiring causal inference and commonsense reasoning.

**Common Sense Reasoning Task**    We consider the WinoGrande dataset (Sakaguchi et al., 2021), which presents a challenging common sense reasoning task. The WinoGrande dataset is designed to be a more difficult and larger-scale version of the original Winograd Schema Challenge, requiring models to demonstrate human-like reasoning capabilities. By including WinoGrande in our experiments, we aim to assess how well our approaches can enhance a model's ability to reason about complex scenarios and make appropriate inferences based on contextual information.

### D.2    OVERVIEW OF PEFT MODULES

In this paper, we mainly consider three types of PEFT modules. In our proposed hybrid-tuning approach, we jointly train one of these PEFT modules with the base LLM to improve the convergence and overall performance. The following paragraphs provide a detailed overview of the three main PEFT modules considered in this study: Prompt Tuning, Prefix Tuning, and Low-Rank Adaptation (LoRA). In our experiments, we follow the default configuration of Zo-Bench code base (Zhang et al., 2024) without making additional modifications. It is worth noting that our hybrid-tuning methods are also applicable to other recently developed PEFT techniques including (1) other LoRA variants such as X-LoRA (Buehler & Buehler, 2024), Llama-Adapter (Zhang et al., 2023b), AdaLoRA (Zhang et al., 2023a), LoHa (Hyeon-Woo et al., 2021), and LoKr (Yeh et al., 2023); (2) other soft prompts techniques such as P-tuning (Liu et al., 2021; 2023); and (3) Infused Adapter by Inhibiting and Amplifying Inner Activation (IA3) methods (Liu et al., 2022).

**Prompt Tuning**    Prompt tuning (Lester et al., 2021) is a lightweight fine-tuning method that prepends trainable continuous prompt tokens to the input. These prompt tokens are optimized during training while keeping the pre-trained language model parameters frozen. This approach allows for task-specific adaptation with a small number of parameters. Prompt tuning is particularly effective for large language models and can be seen as a form of soft prompting that learns optimal input representations for specific tasks.

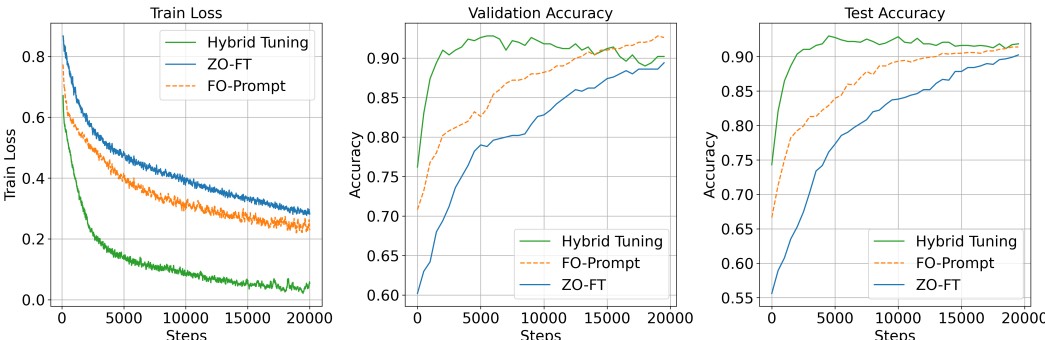

(a) Training curves for OPT-1.3B model with the prompt tuning on the SST2 dataset with using the optimal hyper-parameter indicated in Table 2. The hybrid-tuning achieves the significant better performance. Notably, this phenomenon is also observed in other tasks and for other model architectures.

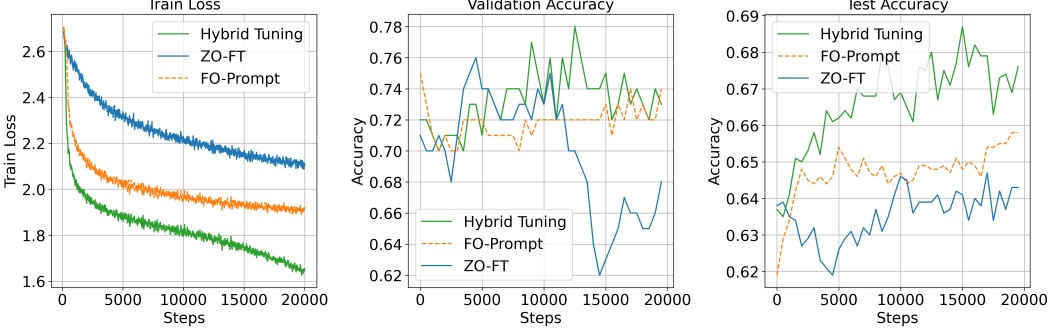

(b) Training curves for Vicuna-7b-v1.5 model with the prompt tuning on the WinoGrande dataset.

Figure 3: Comparison of training curves for different models and datasets. These results demonstrate that the similar outperformance of hybrid-tuning is observed across various model architectures and NLP tasks.

**Prefix Tuning** Prefix tuning (Li & Liang, 2021) extends the concept of prompt tuning by adding trainable prefix tokens not only to the input but to each layer of the transformer model. This method prepends a trainable continuous prefix to the keys and values of the self-attention layers in each transformer block. By doing so, prefix tuning allows for more flexible and expressive task-specific adaptations compared to prompt tuning, while still maintaining a relatively small number of trainable parameters.

**LoRA** Low-Rank Adaptation (LoRA) (Hu et al., 2021) is a parameter-efficient fine-tuning method that adds low-rank decomposition matrices to the weights of the pre-trained model. Instead of directly updating the model's weight matrices, LoRA introduces pairs of rank decomposition matrices for each weight matrix being tuned. These low-rank matrices are initialized randomly and trained to adapt the model to specific tasks. LoRA significantly reduces the number of trainable parameters while maintaining competitive performance compared to full fine-tuning. It offers several advantages, including faster training, lower memory requirements, and the ability to switch between multiple fine-tuned tasks efficiently by changing only the LoRA parameters.

### D.3 CONVERGENCE OF HYBRID FINE-TUNING

In this subsection, we present the training curves (including the training loss, validation accuracy, and the test accuracy) for OPT-1.3B (Zhang et al., 2022) model on SST-2 (Socher et al., 2013) dataset in Figure 3a. We observe that a significant efficiency gain in terms of training steps. The hybrid method consistently achieves optimal performance regarding the training loss. This trend is observed across different tasks, PEFT methods, and model architectures, suggesting that the efficiency of hybrid tuning scales well (e.g. for Vicuna-7b-v1.5 model on the WinoGrande dataset in Figure 3b).

### D.4 ESTIMATING SMOOTHNESS

In Figure 1a and Figure 1b, the smoothness of the loss landscape of the OPT-125M (and the LoRA module) is estimated by approximating the norm of Hessian matrix at the stochastic data point using the zeroth-order gradient estimation to the Hessian-vector products (HVPs):

$$\text{Hessian}(x)^\top v \approx \sum_{\xi \in \text{Batch}} \frac{\nabla f(x + hv; \xi) - \nabla f(x; \xi)}{h},$$

where $\nabla f(x; \xi)$ is the stochastic gradient at $x$ for the data point $\xi$ in the given data batch, $h$ is a small perturbation size, and $v$ is a random unit vector. We estimate the Frobenius norm $\|\text{Hessian}(x)\|_F \approx \sqrt{\mathbf{E} v^\top H^2 v}$ of the Hessian by sampling multiple random vectors and computing these HVPs.

For Figure 1a, we initialize the parameter of pre-trained binary classification OPT-125M model and train it over the SST2 dataset for 5000 steps with setting the learning rate $\eta = 5 \times 10^{-5}$ and the batch size 8. We sample 100 independent vectors from the unit sphere to estimate the HVP with the perturbation $h = 10^{-5}$ and obtain the Hessian norm as the approximation of the local smoothness constant $L$.

For Figure 1b, we initialize the parameter of pre-trained binary classification OPT-125M model as the base model and randomly initialize the LoRA module with the rank 16 and the LoRA Alpha 32 (the detailed configuration can be found in the source code) and jointly train both components over the SST2 dataset for 5000 steps with setting the learning rate $\eta = 5 \times 10^{-5}$ and the batch size 8. We collect all parameters along the SGD trajectories. We perturb the parameter of the base LLM and the LoRA module, respectively, with 100 independent vectors from the unit sphere and the perturbation $h = 10^{-5}$ to estimate the smoothness.

