# OpenReview forum: "Hybrid Fine-Tuning of LLMs: Theoretical Insights on Generalized Smoothness and Convergence"
_ICLR.cc/2025/Conference — ICLR 2025 Conference Withdrawn Submission_

### Official Review · Reviewer_ihvV · 2024-10-30

**Soundness:** 2
**Presentation:** 3
**Contribution:** 1
**Rating:** 3
**Confidence:** 4

**Summary:**

This paper proposes a hybrid fine-tuning approach, that combines regular LLM fine-tuning and PEFT. The proposed method uses first order methods to tune the peft modules, and uses zero-th order methods to tune original parameters. The authors derive convergence rate of the proposed method under the hybrid generalized smoothness assumptions. Empirical results are provided to validate the proposed method.

**Strengths:**

1. The paper is well written and easy to follow.
2. The authors provide convergence analysis as well as empirical results of the proposed algorithm.
3. The observation on learning rate size is interesting to me.

**Weaknesses:**

1. Lack of novelty. The proposed hybrid fine-tuning seems to be a direct combination of zeroth-order algorithms and PEFT algorithms. It is basically running this two algorithms at the same time. The only difference is to use different learning rate for different part of parameters, which seems not novel enough for me, since it is a commonly known fact and well accepted practice that PEFT algorithms requires larger learning rate compared with full parameter fine-tuning. I can provide some references if necessary.

2. The theoretical contribution is not significant. Though this paper spends a lot of work to derive convergence guarantees for the proposed algorithm, the algorithm is actually a variant of SGD with zero-th order noise, with different learning rate for different coordinates, whose convergence guarantee is not fundamentally different from what is well studied in the zero-th order optimization literature. The proposed hybrid generalized smoothness is just a fine-grained version of regular smoothness, by treating different coordinates separately, which I do not think introduces paradigm shift compared with regular smoothness.

3. The empirical results are insufficient. The experiments are conducted on three small scaled downsampled dataset. These datasets are too easy for the pretrained models that have billions of parameters that are used in the experiments. I think the current results are not sufficient to demonstrate the superiority of the proposed algorithm.

**Questions:**

1. What is the point of random shuffling in Algorithm one?

2. Do you also record the memory and computation cost of the hybrid method. It will be interesting to see these results beyond just accuracy comparison.

---

> ### Author Response · Authors · 2024-11-25
>
> Thank you for your detailed review comment and the constructive feedback.
>
> 1. **Novelty**: We would like to emphasis that our observation on using larger learning rate for the PEFT module is based on the theoretical understanding of the hybrid smoothness structure. From our best knowledge, we have not identified the existing literature showing the such hybrid smoothness. We hope the reviewer could kindly provide the related references, which could be an evidence that further validates our theoretical understanding.
> 2. **Theoretical contributions**: Our result presents $O(\epsilon^{-4} + \epsilon^{-2}/\delta)$ sample complexity with the probability $1-\delta$, which improves the existing sample complexity $O(\epsilon^{-4}/\delta)$ of [Li2024]. This improvement is due to our refined proof process with using tighter concentration inequality compared to their original proof. As an example, if we hope to achieve the desired complexity with the probability at least $1-\epsilon^2$, our derived upper bound indicates that it requires at most $O(\epsilon^{-4})$ data samples, which is much sharper than $O(\epsilon^{-6})$ derived from [Li2024].
> 3. **Empirical results**:
>
> * **Question 1**: The point of random shuffling.
>
>   We consider the reshuffling-type SGD because such epoch-wise optimizers are typically more common in machine learning practice
>
> * **Question 2**: The memory and computation cost.
>
>   We appreciate the suggestion of adding the memory and computation cost. We didn't track such cost and we plan to follow this suggestion in the future; the theoretical memory and computation cost are valid in our revised paper.
>
>     | Optimizer                | Theoretical Memory                                   | Asymptotical Memory                       |
>     | ------------------------ | ---------------------------------------------------- | ----------------------------------------- |
>     | FO-SGD (LLM)             | $\sum_\ell \max\\\{  \|a_\ell \|, \|x_\ell\| \\\} + \|x\|$     | $\sum_\ell \max\\\{  \|a_\ell\|, \|x_\ell\| \\\}$ |
>     | Vanilla ZO-SGD (LLM)     | $\|x\|$                                                | $\|x\|$                                     |
>     | FO-SGD (PEFT)            | $\sum_\ell \max\\\{  \|b_\ell \|, \|y\|_\ell \\\} + \|y\|+\|x\|$ | $\|x\|$                                     |
>     | Hybrid ZO-SGD (LLM+PEFT) | $\|y\|+\|x\|$                                          | $\|x\| $                                    |

---

### Official Review · Reviewer_6Th1 · 2024-11-01

**Soundness:** 2
**Presentation:** 3
**Contribution:** 2
**Rating:** 5
**Confidence:** 4

**Summary:**

This work introduces a hybrid fine-tuning paradigm, a novel approach that addresses the limitations of both full fine-tuning and traditional parameter-efficient fine-tuning (PEFT) methods. By integrating zero-order optimization for large language models (LLMs) with first-order optimization for PEFT modules, this method achieves an effective balance between adaptability and computational efficiency.

**Strengths:**

The novel hybrid generalized smoothing concept expands classical optimization theory to account for the heterogeneous dynamics of joint training between large language models (LLMs) and parameter-efficient fine-tuning (PEFT) methods. This approach is versatile, applicable to hybrid fine-tuning, layer-wise fine-tuning, and models incorporating trainable external modules.

**Weaknesses:**

My primary concern lies with the performance of the proposed method. In the experiments, it does not significantly surpass the baseline methods. Additionally, beyond the vanilla zeroth-order SGD, other advanced zeroth-order methods are available, as discussed in [1]. I suggest that the authors incorporate these alternative methods as baselines to further validate the effectiveness of their approach.

[1] Zhang, Yihua, et al. "Revisiting zeroth-order optimization for memory-efficient llm fine-tuning: A benchmark." arXiv preprint arXiv:2402.11592 (2024).

**Questions:**

1. Why is it not feasible to train each module sequentially with distinct learning rates? Are there any specific benefits to mixing them within the same training phase?
2. In the second part of Figure 3 in Appendix D.2, why does the accuracy rate of hybrid tuning show a decreasing trend over time steps?

---

> ### Author Response · Authors · 2024-11-25
>
> Thank you for your detailed review comment and the constructive feedback.
>
> **Question 1**: Is it not feasible to train each module sequentially?
>
> Yes, sequentially updating each module is an alternative approach. However, it will double the time cost for each batch of data since it requires two rounds of forward + backward passes.
>
> **Question 2**:  The accuracy rate of hybrid tuning decreases.
>
> Adding the PEFT module to the base model introduces additional parameters; therefore, the model is more likely to be overfitting. The validation loss exactly validate this phenomenon. The reason why this phenomenon only appears to the hybrid tuning method is that the hybrid learning rate significantly accelerates the training procedure, which is also evidenced by the training loss of Figure 3.

---

### Official Review · Reviewer_rzJC · 2024-11-02

**Soundness:** 3
**Presentation:** 3
**Contribution:** 3
**Rating:** 5
**Confidence:** 3

**Summary:**

Applying either Parameter-Efficient Fine-Tuning (PEFT) or full fine-tuning to Large Language Models (LLMs) often results in its inherent limitations. To overcome this issue, this paper proposes a novel "hybrid fine-tuning" approach that jointly updates both LLMs and PEFT modules using a combination of zeroth-order and first-order optimization methods.

**Strengths:**

Applying either Parameter-Efficient Fine-Tuning (PEFT) or full fine-tuning to Large Language Models (LLMs) often results in its inherent limitations. To overcome this issue, this paper proposes a novel "hybrid fine-tuning" approach that jointly updates both LLMs and PEFT modules using a combination of zeroth-order and first-order optimization methods.

**Weaknesses:**

The combination of PEFT and full fine-tuning seems to be a trivial trick.
The core of this paper is using zeroth-order algorithm to achieve full fine-tuning and Adam to PEFT.
Though this is an effective method, this combination seems so trivial.

**Questions:**

No

---

> ### Author Response · Authors · 2024-11-25
>
> Thanks for your comment. The combination is indeed a trivial trick; however, our main contribution also identifies the hybrid smoothness of its inherent structure, which presents the theoretical foundations on why it is more effective to set different learning rates.

---

> > ### Comment · Reviewer_rzJC · 2024-11-29
> >
> > Thank you for your replies. However, the idea that using different steps for different coordinates to achieve faster convergence rate is not new, either.

---

> > > ### Author Response · Authors · 2024-11-30
> > >
> > > Thanks for the reply. Our main contribution also identifies the hybrid smoothness of its inherent structure, which presents the theoretical foundations on why it is more effective to set different learning rates and it has not been addressed in other places. We sincerely hope you consider this point as the novelty. We also note that we have never claimed that using different learning rates is a new design in our paper.

---

### Official Review · Reviewer_HxaN · 2024-11-03

**Soundness:** 1
**Presentation:** 1
**Contribution:** 1
**Rating:** 3
**Confidence:** 5

**Summary:**

The paper proposes to combine a zero-order method with a parameter-efficient fine-tuning technique for LLM fine-tuning. Experiments test the effectiveness. The paper is not well written.  There is no technical and theoretical contributions.


----
I appreciate the authors' response, but the paper still requires further conciseness and the necessary experiments. I raise my score, but currently, the paper cannot reach the threshold of ICLR.

**Strengths:**

- Combining a zero-order method with a parameter-efficient fine-tuning technique for LLM fine-tuning is a feasible approach.

- The convergence of the proposed method is discussed in this paper.

- This paper test the effectiveness of the proposed method on small datasets.

**Weaknesses:**

- The paper lacks technical contributions. The paper combines a zero-order method with a parameter-efficient fine-tuning technique for LLM fine-tuning. However, it provides no specific details on how to integrate the network weight updates from both approaches and lacks analysis or testing. For example, one obtains $\Delta W_1$ and $\Delta W_2$ to update the network weight $W$ by the zero-order method and a parameter-efficient fine-tuning method, respectively. How to update $W$ with $\Delta W_1$ and $\Delta W_2$ in the end? The only distinction made is that each approach uses a different learning rate, which does not enhance the hybrid methodology.

- The paper fails to offer theoretical contributions, as most lemmas and theorems are derivable from (Li et al., 2024). The paper does not clarify why its proofs and conclusions cannot be directly applied for theoretical analysis or what challenges arise from its application. There is no theoretical contribution if only the processes of proofs are slightly different.

Haochuan Li, Jian Qian, Yi Tian, Alexander Rakhlin, and Ali Jadbabaie. Convex and non-convex
optimization under generalized smoothness. Advances in Neural Information Processing Systems,
36, 2024.

- Additionally, the experiments are weak. The experimental setup, comparison methods, and discussion of results indicate inadequate training of the authors in scientific research. It is a reasonable choice to directly repeat the experiments in MeZO.

**Questions:**

- This paper proposes a hybrid approach combining zero-order full parameter fine-tuning with first-order parameter-efficient fine-tuning. So the comparison of memory and wall-clock time is very important. However, it lacks experimental verification and discussion.

- The experiments lack strength, and the datasets are limited in size and quantity. Given that MeZO's experimental setup is utilized, it would be beneficial to fellow its experiments.

- It should give the results of the first-order methods SGD or Adam as references.

- The methods in Table 1 do not align with those discussed in the experimental details. Please either cite the relevant papers in Table 1 or include the corresponding methods in the experimental details subsection.

- The statement above Eq. (2) is incorrect. The equations of one-side and two-side gradient estimators are different and cannot be expressed by one equation.

- The zero-order optimizer used in all experiments needs to be explained. The first-order optimizer used in parameter-efficient fine-tuning also needs to be explained.

---

> ### Author Response · Authors · 2024-11-25
>
> Thank you for your review and feedback. We add the following point-to-point responses. Due to the limited time of rebuttal phase, we didn't add additional experiments. However, we sincerely hope the reviewer could value our novelty in the theoretical analysis as discussed below.
>
> **Difference from [Li2024]**: We would like to highlight the main theoretical difference between our analysis and [Li2024]: Our result presents $O(\epsilon^{-4} + \epsilon^{-2}/\delta)$ sample complexity with the probability $1-\delta$, which improves the existing sample complexity $O(\epsilon^{-4}/\delta)$ of [Li2024]. This improvement is due to our refined proof process with using tighter concentration inequality compared to their original proof. As an example, if we hope to achieve the desired complexity with the probability at least $1-\epsilon^2$, our derived upper bound indicates that it requires at most $O(\epsilon^{-4})$ data samples, which is much sharper than $O(\epsilon^{-6})$ derived from [Li2024].
>
> **Lacks analysis or testing**: We have provided theoretical analysis with the convergence guarantee and empirically verified the performance of our proposed method. We have clearly stated the methodology in our paper: Our method is simply applying different learning rates, which is motivated by the inherent hybrid structure of PEFT method.
>
> **Experiments are weak**: We have tested our methods over three representative datasets: Test Classification, Question Answering, and Common Sense Reasoning. Adding more datasets in the same category won't enhance our statement.
>
> Here, we answer the question raised by the reviewer:
>
> * **Question 1**: Comparison of memory and wall-clock time.
>
>   We appreciate the suggestion of adding the comparison of memory and wall-clock time. We have added a theoretical memory and wall-clock time of our proposed method as follows:
>
>   * **Memory Cost**: We summarize the memory consumption comparison in the following table. The memory consumption of the PEFT model’s parameters is commonly significantly smaller than the base language model's parameters. Therefore, in the asymptotic analysis, we can omit its impact on our memory estimation.
>
>     | Optimizer                | Theoretical Memory                                   | Asymptotical Memory                       |
>     | ------------------------ | ---------------------------------------------------- | ----------------------------------------- |
>     | FO-SGD (LLM)             | $\sum_\ell \max\\\{  \|a_\ell \|, \|x_\ell\| \\\} + \|x\|$     | $\sum_\ell \max\\\{  \|a_\ell\|, \|x_\ell\| \\\}$ |
>     | Vanilla ZO-SGD (LLM)     | $\|x\|$                                                | $\|x\|$                                     |
>     | FO-SGD (PEFT)            | $\sum_\ell \max\\\{  \|b_\ell \|, \|y\|_\ell \\\} + \|y\|+\|x\|$ | $\|x\|$                                     |
>     | Hybrid ZO-SGD (LLM+PEFT) | $\|y\|+\|x\|$                                          | $\|x\| $                                    |
>
>     Where $a_\ell, b_\ell$  represents the total activations being stored for computing the backward gradients and $x_\ell, y_\ell$ represents the number of parameters in $\ell$-th layer of the base LLM model and the PEFT model, respectively.
>
>   * **Wall-clock time**: Our method doesn't involve any modification on the original backward and forward step. Therefore, our wall-clock time is the same as the classical zeroth-order method reported in MeZo and Zo-Bench papers.
>
> * **Question 2 \& 3 \& 4**: Follow the experiments given in MeZo \& First-order SGD and Adam baseline \& Revise Table 1 to add references.
>
>   We appreciate this suggestion. We will add these experiments in the future.
>
> * **Question 5**: Statement above Eq.(2) is incorrect.
>
>   Thanks for pointing it out. We will fix this typo.
>
> * **Question 6**: Optimizers used need to be explained.
>
>   We use the vanilla SGD without momentum and weight decaying for both LLM and PEFT modules. We will make it more clear in our future submission.

---

### Author Response · Authors · 2024-12-02

Dear Reviewers,

We sincerely appreciate your thorough reviews and insightful comments. If you have any further questions or feedback, please do not hesitate to let us know. We are more than willing to engage in the discussion and further improve our work.

Sincerely,

The Authors

---

### Note · Authors · 2025-01-22

I have read and agree with the venue's withdrawal policy on behalf of myself and my co-authors.